# Functional-metabolic coupling in distinct renal cell types coordinates organ-wide physiology and delays premature ageing

Jack Holcombe [1] & Helen Weavers [1]✉

Precise coupling between cellular physiology and metabolism is emerging as a vital relationship underpinning tissue health and longevity. Nevertheless, functional-metabolic coupling within heterogenous microenvironments in vivo remains poorly understood due to tissue complexity and metabolic plasticity. Here, we establish the *Drosophila* renal system as a paradigm for linking mechanistic analysis of metabolism, at single-cell resolution, to organ-wide physiology. Kidneys are amongst the most energetically-demanding organs, yet exactly how individual cell types fine-tune metabolism to meet their diverse, unique physiologies over the life-course remains unclear. Integrating live-imaging of metabolite and organelle dynamics with spatio-temporal genetic perturbation within intact functional tissue, we uncover distinct cellular metabolic signatures essential to support renal physiology and healthy ageing. Cell type-specific programming of glucose handling, PPP-mediated glutathione regeneration and FA β-oxidation via dynamic lipid-peroxisomal networks, downstream of differential ERR receptor activity, precisely match cellular energetic demands whilst limiting damage and premature senescence; however, their dramatic dysregulation may underlie age-related renal dysfunction.

Metabolic networks are immensely complex, with increasing numbers of metabolites playing key roles beyond cellular energetics and ATP generation. These networks are also highly dynamic and diverse. It is well-known that metabolic signatures within individual cells change over their lifetime as they differentiate, age or encounter new challenges[1–3]; for example, individual cells may adapt their metabolism as substrate availability changes or they encounter low oxygen conditions. Nevertheless within complex tissues - where heterogenous populations of diverse cell types frequently co-exist – individual constituent cell types are also expected to fine-tune their metabolic profiles to match their unique physiological roles. These distinct cellular metabolic and functional specialisations must in turn act holistically to support the overall function and survival of the tissue[4]. Such spatio-temporal metabolic compartmentalisation may have arisen from an evolutionary pressure to reduce competition for precious substrates[1–3]. Nevertheless, our current understanding of metabolic heterogeneity among cells within complex tissues in vivo over the life-course, and how this relates to their specific physiological roles, has been limited by challenges associated with metabolic plasticity and tissue complexity.

The kidney is a paradigm tissue to dissect how cellular metabolism is fine-tuned within complex heterogenous microenvironments to support key cellular processes and tissue-wide physiology. Kidneys play vital roles in balancing our internal environment, through precise osmoregulation and excretion of metabolic (or potentially toxic) waste. Overall renal function relies on ATP-dependent processes, particularly the ability to perform extensive active transport (e.g., via the $Na^+$ $K^+$ ATPase) for resorption of solutes, amino acids and other essential molecules back into the blood[5]. Kidneys consequently have an extraordinarily high bioenergetic demand, with an estimated metabolic rate of 440 kcal per kg per day (in contrast to 200 kcal, 240 kcal, 13 kcal and 4 kcal per kg per day for the liver, brain, skeletal

[1]School of Biochemistry, Biomedical Sciences, University of Bristol, Bristol BS8 1TD, UK. ✉e-mail: helen.weavers@bristol.ac.uk

muscle and adipose tissue, respectively)[6]. Renal ATP production is so tightly coupled to consumption by the renal ATPase, ATPase activity appears to serve as a pacemaker for cellular respiration[7].

Nevertheless, the mammalian kidney is a complex heterogenous tissue, with multiple cell types each with distinct physiological characteristics and bioenergetic demands. Although the kidneys main regions are known to exhibit broadly different metabolic profiles - with the renal medulla relying predominantly upon anaerobic glycolysis for energy, whilst the mitochondria-rich renal cortex has abundant oxidative enzymes and little glucose-phosphorylating capacity[8] - this metabolic heterogeneity likely occurs over far smaller scales even between directly opposed cells. Indeed, the recent surge in single-cell 'omics has revealed cell-type specific differences in metabolites and metabolic gene expression in healthy and diseased kidneys[9]. However, the functional relevance of this spatially-resolved metabolism for renal physiology remains unclear due to challenges associated with cell type-specific mechanistic studies in vivo. There is a pressing need to understand the precise metabolic profiles employed by different renal cellular subtypes and how these heterogenous profiles are coordinated in space and time to sustain organ-wide function.

Since cells reside in specialised tissue microenvironments that precisely regulate their biology, metabolic networks are best studied in their native, in vivo context. The study of cellular metabolism, and its wider role in organ physiology, has been traditionally challenging in mammalian models due to tissue complexity and the plasticity of metabolic networks, as well as a shortage of experimental tools to live image and precisely manipulate the metabolism of individual cells within the intact, living functional tissue. Much previous mechanistic work on metabolism has provided relatively crude spatial resolution or assessed metabolic states of isolated cell populations in vitro. Given the dynamic and sensitive nature of metabolic pathways, loss of tissue integrity may drastically alter the metabolic configuration of cells. Recent work has established the fruit fly as a powerful platform for dissecting complex metabolic relationships within and between organs in vivo (such as the reproductive organs and gut)[10–14] using superlative experimental tools as-yet unavailable in mammalian systems. Here, we exploit *Drosophila's* unrivalled experimental tractability to perform comprehensive mechanistic analysis of cellular metabolism in functionally divergent (yet physically-coupled) renal cell types within their intact, living microenvironment in vivo. We dissect how these distinct, spatially-resolved metabolic profiles sustain subcellular processes, limit cellular injury and support robust organ-wide physiology to delay premature ageing.

The *Drosophila* renal system is well-established as a model to study fundamental aspects of renal development, cell biology and physiology[15,16]. As in the mammalian nephron, the structure and function of the insect renal tubule is highly compartmentalised, with broad functional divergence along the tubule's length[17], as well as distinct cell types each with unique physiological roles. Whilst the Principal cells (PCs) perform metabolically-intense active cation transport, the directly apposed Stellate cells (SCs) offer a 'privileged route' for more passive chloride and water conductance[16,18]. Recent comparison of *Drosophila* and mouse single cell transcriptomic data suggest that PCs are analogous to mammalian proximal tubules whilst SCs are analogous to the (highly water permeable) thin limb of the mammalian lower loop of Henle[19]. Insect renal tubules are the fastest fluid-secreting epithelia known in biology[20] and thus offer a uniquely powerful system to determine how spatially-resolved metabolism supports a tissue's intense physiological demands. *Drosophila* renal tubules are also unique in being one of the few differentiated tissues that persist from *Drosophila* embryogenesis through to adulthood, with most other tissues being extensively remodelled during pupation[21]; this positions the insect renal tubule as a particularly vital tissue.

In this study, we demonstrate that strategic metabolic partitioning between distinct renal cell types not only supports their unique functional roles but also sustains tissue-wide excretory activity and delays premature ageing. We integrate live, quantitative time-lapse imaging of metabolite and organelle dynamics (at subcellular resolution) and cell-type specific genetic perturbation over the life-course with elegant organ-wide physiological assays, all within intact, living renal tubules. Although renal activity is tightly linked to mitochondrial ATP generation (as excretion arrests within minutes of pharmacological Electron Transport Chain blockade), we find robust excretion critically depends on the adoption of distinct metabolic profiles by different constituent cell types. Active transport-dependent PCs rely chiefly upon fatty acid (FA) mobilisation via a dynamic peroxisomal-lipid droplet storage network to maximise energy production through FA oxidation (FAO)-mediated ATP generation, enabling carbohydrate diversion for NADPH and glutathione regeneration to neutralise mitochondrial ROS and limit premature senescence. In contrast, directly apposed renal SCs exhibit markedly distinct metabolism, independent of FAO and pentose phosphate pathway (PPP) activity, with their minimal ATP demand likely met by glucose flux through glycolysis and oxidative phosphorylation (OXPHOS). This strict functional-metabolic programming is coordinated by differential activity of the conserved nuclear receptor Estrogen-related Receptor (ERR). Cell-type specific programming of complex intersecting metabolic networks thus ensures individual cellular energetic, biosynthetic and physiological demands are precisely met whilst limiting renal injury and premature ageing, and this in turn holistically supports whole organ function. Whilst these cell type-specific metabolic adaptations critically support renal function in young heathy individuals, their marked disruption with age associates with declining organ-wide excretion. We hypothesise that such precise metabolic programming, particularly its spatial compartmentalisation, could have evolved to limit competition for scarce resources and ensure precious metabolites are partitioned most appropriately within a single tissue. Strict metabolic partitioning may be especially crucial given that excessive levels of even protective metabolites (e.g., glutathione) can be harmful[22].

A better understanding of how complex organs like the kidney fine-tune their metabolism will undoubtedly aid the discovery of new therapeutic targets to combat degenerative disease and prolong healthy ageing. Indeed, increasing evidence suggests metabolic dysregulation may contribute to numerous renal pathologies, including acute kidney injury (AKI), chronic kidney disease (CKD) and age-related renal decline[23]. Renal transcriptomics has revealed that metabolic genes, such as those involved in lipid metabolism and OXPHOS show a strong correlation with CKD in patients and mouse models[24,25]. Moreover, recent 'omics profiling of donor biopsies showed age-related changes in certain metabolic proteins could predict renal dysfunction post-transplant[23]. We envision the *Drosophila* renal tubules will offer a powerful, experimentally-tractable in vivo platform to identify novel therapeutic entry points to target the complex metabolic relationships that underpin healthy renal ageing.

## Results

### Preferential carbohydrate import and early metabolism in PCs (but not SCs) supports renal physiology independent of ATP production

*Drosophila* renal (Malpighian, MpT) tubules consist of two major cell types, the larger PCs and smaller SCs which perform distinct roles in ion, solute and water transport (Fig. 1a, b)[16]. These functional differences are reflected in their distinct expression profiles from recent single nuclear transcriptomic studies (Fig. S1a Fly Cell Atlas data[19,26,27]). Like the mammalian nephron, active transport is primarily driven by an ATP-dependent ATPase (the V-ATPase proton pump) whose activity is spatially restricted to PCs (Fig. 1b)[16]. PCs

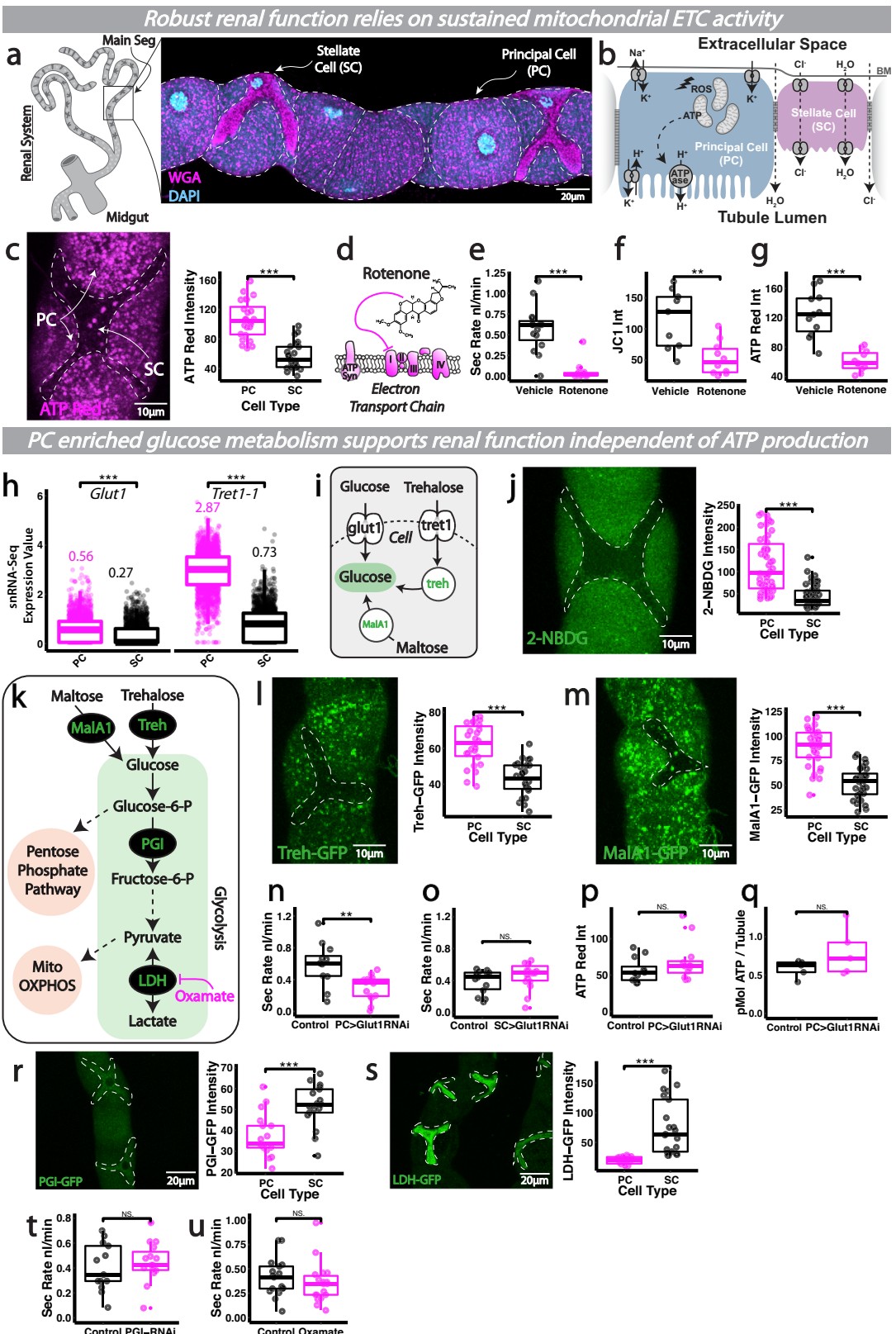

thus have huge energy demands (Fig. 1c, ATP Red) and are dramatically enriched in mitochondria (Fig. S1b, c)[28,29] compared to their SC neighbours, the latter performing more passive roles in water and chloride movement[18,21,30]. Blockade of V-ATPase with bafilomycin nearly doubles cellular [ATP], confirming the V-ATPase as the largest single ATP sink in the tubule[29]. Consistent with this, PCs exhibit significantly higher expression of OXPHOS related genes

(Fig. S1d Fly Cell Atlas data[19,26]) and have higher mitochondrial activity (as shown using the voltage-sensitive dye JC-1 that assess mitochondrial potential, Fig. S1e) than their SC neighbours[29]. Tubule excretory function is absolutely dependent on sustained ATPase activity, as well as robust mitochondrial ATP production, as pharmacological inhibition of ATPase[31] or rotenone-driven inhibition of mitochondrial electron transport (targeting complex I)

**Fig. 1 | Functionally diverse renal subtypes exhibit distinct metabolic profiles in vivo. a** Schematic and confocal image of *Drosophila* renal tubules showing the two major cell types, the principal cells and stellate cells (**b**). The mitochondria-rich principal cell actively transports protons via an apical vacuolar H$^+$-ATPase (V-ATPase), setting up a gradient which is exchanged primarily for potassium which enters the cell basolaterally through the combined activity of Na$^+$, K$^+$-ATPase, inward rectifier potassium channels and potassium cotransporters. **c** Analysis and quantification of renal tubule ATP-Red staining. Mitochondrial ETC inhibition via rotenone treatment (schematic, **d**), its impact on renal secretory activity (**e**), mitochondrial activity (JC-1, **f**) and bioenergetic output (ATP-Red, **g**). **h** Expression of glucose and trehalose transporters from publicly available snRNA-Seq data (Fly Cell Atlas). **i** Schematic of glucose/trehalose import and derivation of glucose from more complex sugars. **j** Analysis and quantification of 2-NBDG glucose import in SCs and PCs. **k** Schematic of glucose metabolism and glycolytic inhibitor action (Oxamate). Analysis and quantification of Treh-GFP (**l**) and MalA1-GFP (**m**) mean fluorescent intensity in renal tubules. **n–q** Analysis and quantification of renal secretory activity and bioenergetic output in *Glut1-RNAi* tubules, driven in PCs (*CapaR-Gal4*) or SCs (*C724-Gal4*). Analysis and quantification of PGI-GFP (**r**) and LDH-GFP mean fluorescent intensity (**s**) in renal tubules. Quantification of renal secretory activity when glycolysis is inhibited via *CapaR>PGI-RNAi* and oxamate

treatment (**t**, **u**). Data represented as box and whisker plots (lower and upper hinges correspond to the first and third quartiles, median line within the box, whiskers extend from the hinge to the largest/smallest value, at most 1.5* interquartile range of the hinge) with all data from MpT cells (SCs or PCs, **c**, **j**, **l**, **m**, **r**, **s**), MpT main segment sections (**f**, **g**, **p**, **v**) or secretion of individual kidneys (**e**, **n**, **o**, **s**, **t**) shown as overlaid points. NS Not Significant, *$P < 0.05$, **$P < 0.01$, ***$P < 0.001$ (unpaired two-tailed *t*-tests or Wilcoxon test with FDR correction). *p* values: **c** $p < 0.0001$, **e** $p < 0.0001$, **f** $p = 0.00395$, **g** $p < 0.0001$, **h** Glut1 $p < 0.0001$, Tret1-1 $p < 0.0001$, **j** $p < 0.0001$, **l** $p < 0.0001$, **m** $p < 0.0001$, **n** $p = 0.00408$, **r** $p = 0.000288$, **s** $p < 0.0001$. *p* values where $p > 0.05$ labelled as NS. For analysis of fluorescent reporters/dyes, two images of different sections of the MpT main segment per fly were imaged. All images are maximum z projections. **c** $n = 24$ cells (from 7 tubules) per condition, **e** $n = 15$ tubules per condition, **f** $n = 9$ vehicle and 10 rotenone tubules, **g** $n = 11$ tubules per condition, **j** $n = 50$ cells (from 8 tubules) per condition, **l** $n = 26$ cells (from 6 tubules) per condition, **m** $n = 32$ cells (from 9 tubule) per condition, **n** $n = 12$ control and 14 RNAi tubules, **o** $n = 14$ control and 18 RNAi tubules, **p** $n = 12$ tubules per condition, **q** $n = 5$ extracts (30 tubules per extract), **r** $n = 17$ cells (from 5 tubules) per condition, **s** $n = 23$ cells (from 3 tubules), **t** $n = 13$ control and 15 RNAi tubules, **u** $n = 17$ control and 18 oxamate tubules. Source data are provided as a Source Data file.

abolished the secretory capacity of intact, living renal tubules (Figs. 1d–g and S1f).

Given that renal activity is tightly linked to ATP supply, modest changes in energy metabolism may have profound effects on tubule function. We speculated that PCs, in particular, may have to optimise energy production to meet their intense ATP demands, and explored whether PCs and SCs exhibited metabolic differences, including their ability to import (and metabolise) major energy sources. PCs expressed higher levels of transporters for both glucose (*glut1*) and trehalose (the major circulating sugar in *Drosophila*[32]) (*tret-1-1*) than SCs (Fig. 1h, i, Fly Cell Atlas single nucleus transcriptomic data[26]). Crucially, live-imaging of 2-NBDG (a fluorescent glucose analogue) uptake in individual PCs within intact, living renal tubules revealed significantly greater uptake by PCs relative to adjacent SCs (Fig. 1j). PCs also expressed higher levels of GFP reporters for the enzymes maltase and trehalase that derive glucose from maltose and trehalose (Fig. 1k–m); while it cannot be asserted that any individual reporter construct will exactly represent the expression of its cognate gene, the differential levels of Treh-GFP and MalA1-GFP suggest PCs express higher levels of these enzymes than SCs. Together, these data suggest PCs may specifically programme their metabolism for heightened ability to import glucose (and to derive it from more complex sugars). Consistent with this, glucose import by PCs was essential for tubule-wide function (Fig. 1n) as *glut1-RNAi* targeted specifically to PCs (using *CapaR-gal4*) significantly perturbed the excretory activity of intact, living renal tubules. In contrast, SC-specific expression of the same *glut1-RNAi* construct (using a SC-specific driver *C724-gal4*, Fig. S1g) had little effect on tubule function under basal conditions (Fig. 1o), suggesting that glucose import by SCs might not be essential for tubule function. PC and SC *Gal4* drivers achieved equivalent levels of target gene expression as shown by *UAS*-driven GFP expression (Fig. S1h). Consistent with these data, expression of the same *glut1-RNAi* construct throughout the tubule (using the ubiquitous driver *Act5c-gal4*) did not reduce tubule secretory capacity further than that observed using PC-specific *CapaR-gal4* (Fig. S1i). Intriguingly, mammalian V-ATPase trafficking, assembly and activity is proportional to glucose concentration and early glycolytic flux[33,34]; perhaps the extensive glucose flux through early glycolysis in PCs (but not SCs) helps maintain robust V ATPase activity.

PC-specific *glut1-RNAi* did not perturb ATP generation within these cells (Fig. 1p, q), despite the associated dramatic effects on tubule excretion, suggesting that PCs may not rely upon imported glucose to support mitochondrial energy metabolism. Indeed,

expression analysis of key enzymes in the glycolytic pathway (Fig. 1k) revealed that PCs exhibited significantly lower density of GFP reporters for Phosphoglucose Isomerase (PGI) and Lactate Dehydrogenase (LDH) than their SC neighbours (Fig. 1r, s); nevertheless given PCs are significantly larger than SCs, our data suggest total levels of PGI (as measured by total GFP fluorescence) are higher in PCs than SCs, although total LDH levels do not differ between cell types (Fig. S1j, k). While it cannot be asserted that any individual reporter construct will completely and faithfully map the expression of its cognate gene, the differential densities of PGI-GFP and LDH-GFP reporters suggest PCs might possess overall lower cytosolic concentrations of late-stage glycolytic enzymes than SCs. Moreover, genetic-mediated (via *PGI-RNAi*) or pharmacological inhibition of glycolysis via Oxamate treatment (a known LDH inhibitor[35]) did not perturb tubule-wide secretory activity (Fig. 1t, u); in contrast to marked inhibitory effects of *PGI-RNAi* and Oxamate (equivalent treatment) on glycolytic readouts (lactate production) within enterocytes (Fig. S1l, m), suggesting that glycolysis is not crucial for homeostatic energy production in renal tubules.

## Lipid metabolism supports renal PCs intense bioenergetic demands and organ-wide physiology

If PCs do not rely predominantly on glucose flux through lower glycolysis to directly meet their energetic demands, how do PCs fuel their physiologically-demanding roles within the kidney? Live imaging of PCs within intact renal tubules revealed PCs import significantly more lipids (Fig. 2a) and possess far more lipid storage droplets (Fig. 2b) than their SC neighbours, suggesting that PCs may supplement their huge energy requirements via lipid metabolism. Consistent with this, PCs exhibit higher expression of key enzymes in fatty acid metabolism and β-oxidation than SCs (Fig. S2a, Fly Cell Atlas data[26]). FAO (fatty acid oxidation) may be a preferred energy source as this pathway produces over 3 times the amount of energy as glycolysis; indeed, the complete oxidation of 1 mole of palmitate (a common saturated fatty acid) results in 130 mole ATP, while oxidation of 1 mole glucose produces 38 mole ATP[36].

To explore the role of mitochondrial lipid metabolism in PCs, we inhibited the major rate-limiting step in mitochondrial FAO (mitochondrial FA import, Fig. 2c) specifically in PCs via RNAi-mediated knockdown of carnitine O-palmitoyltransferase (dCPT1, *Dmel withered*), which is enriched in PCs (Fig. S2a). Following inhibition of the *dCPT1*-dependent mitochondrial carnitine shuttle, individual PCs dramatically accumulated lipid droplets (Fig. 2d–h and Fig. S2b), suggesting that lipids were backing up in the cytosol (a phenomenon known as 'backwards failure'). This was accompanied by a decline in

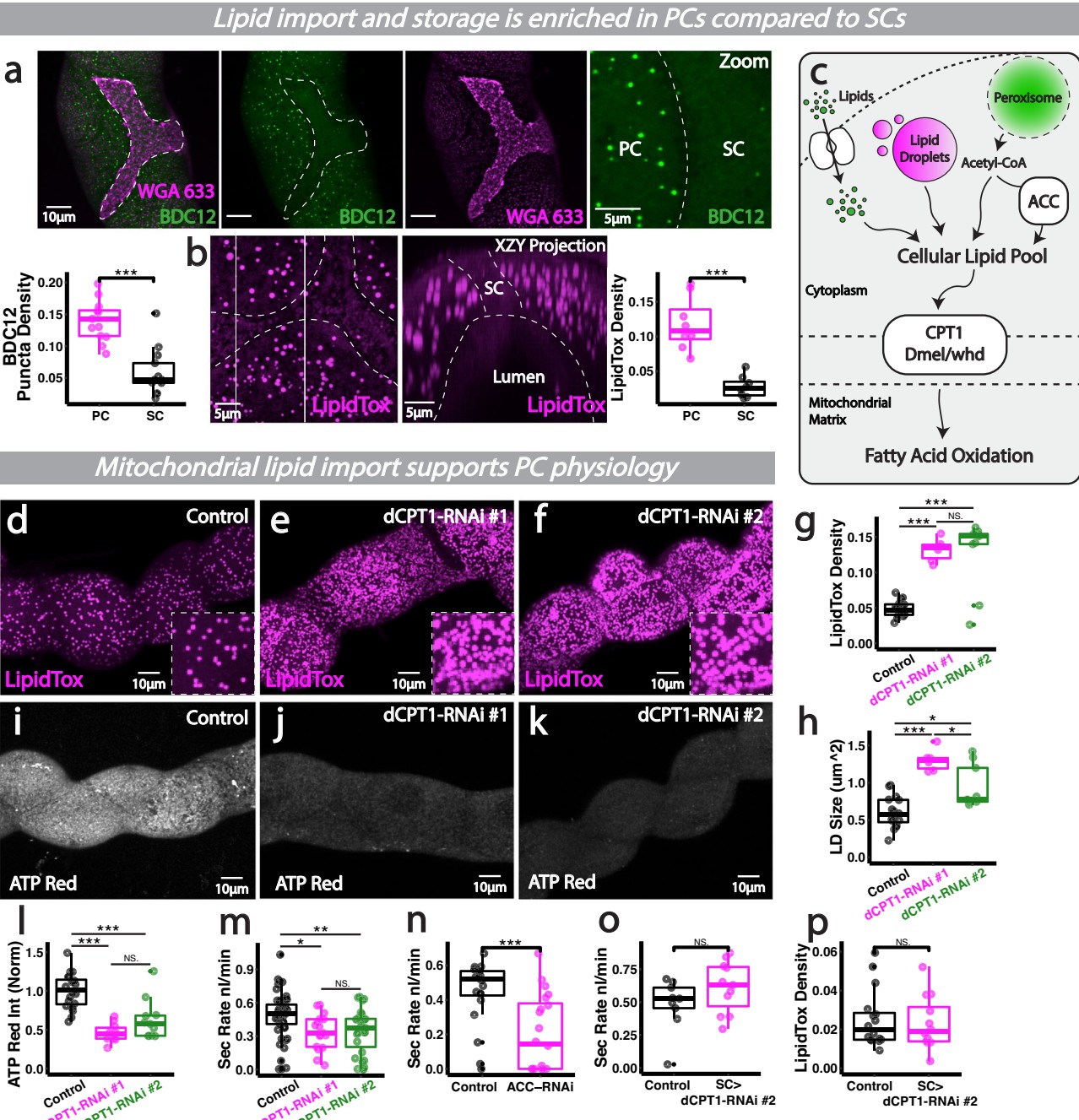

**Fig. 2 | Lipid metabolism supports PCs intense bioenergetic demands and organ-wide function. a** Analysis and quantification of lipid import (BODIPY C12, WGA633 shows cell membrane) and lipid droplets (**b**), LipidTox in renal tubules. **c** Schematic of sources contributing to the cellular lipid pool. **d–h** Analysis and quantification of lipid droplet density/size and bioenergetic output (**i–l**) and secretory activity (**m**) in *CapaR>CPT1-RNAi* (*whd*) tubules. *CPT1-RNAi#1* (V#105400); *CPT1-RNAi#2* (B#34066). **n** Analysis and quantification of secretory activity in *CapaR>ACC-RNAi* renal tubules. **o, p** Analysis and quantification of secretory activity and SC lipidtox density in *CPT1-RNAi* driven in SCs (*C724>CPT1-RNAi*). Data represented as box and whisker plots (lower and upper hinges correspond to the first and third quartiles, median line within the box, whiskers extend from the hinge to the largest/smallest value, at most 1.5* interquartile range of the hinge) with all data from MpT cells (SCs or PCs, **a**, **b**, **p**), MpT main segment sections (**g**, **h**, **l**) or secretion of individual kidneys (**m**, **n**, **o**) shown as overlaid points. NS Not Significant, *$P < 0.05$, **$P < 0.01$, ***$P < 0.001$ (unpaired two-tailed *t*-tests or ANOVA followed by Tukey multiple comparisons test where appropriate). *p* values: (**a**) $p < 0.0001$, (**b**) $p = 0.000156$, (**g**) control vs #1 $p < 0.0001$, control vs #2 $p < 0.0001$, (**h**) control vs #1 $p < 0.0001$, control vs #2 $p = 0.00294$, #1 vs #2 $p = 0.0158$, (**l**) control vs #1 $p < 0.0001$, control vs #2 $p = 0.000138$, (**m**) control vs #1 $p = 0.0283$, control vs #2 $p = 0.00841$, (**n**) $p = 0.000639$. *p* values where $p > 0.05$ labelled as NS. For analysis of fluorescent reporters/dyes, two images of different sections of the MpT main segment per fly were imaged. All images representative of >5 tubules. All images are maximum z projections, aside from B ('XZY Projection') which is a re-sliced max projected transverse section. (**a**) $n = 42$ cells (from 13 tubules) per condition, (**b**) $n = 14$ cells (from 8 tubules) per condition, (**g**, **h**) $n = 16$ control, 6 #1 and 9 #2 tubules, (**l**) $n = 20$ control, 10 #1 and 11 #2 tubules, (**m**) $n = 49$ control, 17 #1 and 31 #2 tubules, (**n**) $n = 20$ tubules per condition, (**o**) $n = 9$ control and 11 RNAi tubules, (**p**) $n = 16$ cells (from 5 tubules) per condition. Source data are provided as a Source Data file.

ATP levels (Fig. 2i–l and Fig. S2c) and significantly reduced tubule excretion (Fig. 2m), suggesting that lipid metabolism is key for PC energy supply and organ-wide physiology. Similarly, RNAi-mediated knockdown of *Drosophila* ACC (Acetyl-CoA carboxylase), the rate limiting step in fatty acid synthesis from acetyl coA, specifically in PCs also perturbed tubule secretion (Fig. 2n), reinforcing the importance of fatty acid metabolism in sustaining renal health. In contrast, inhibition of fatty acid metabolism specifically in SCs (using C724 > *dCPT1-RNAi*) did not perturb tubule excretion (Fig. 2o) nor lead to an accumulation of lipid droplets in the cytosol (Fig. 2p), suggesting SCs (unlike PCs) do not heavily rely upon mitochondrial FAO to meet their energetic demands.

Interestingly, cellular lipid stores have recently been linked to conferring resistance to oxidative stress[37]; the striking enrichment of lipid droplets within PCs could thus offer benefits beyond energy generation, particularly as PCs are vulnerable to oxidative damage due to their intense mitochondrial activity[28]. Nevertheless, at high concentrations free FAs may directly inhibit tubule physiology via disturbing the renal ATPase or mitochondrial membrane potential[38]. It may thus be critical that PCs strike a fine balance in maintaining sufficient cytosolic FAs to support efficient energy production and stress resistance, whilst not impinging upon renal transport activity via accumulating (potentially renotoxic) FA levels.

## PC peroxisomes dynamically interact with cellular lipid stores and support renal function

Given the reliance of PCs on FAO to support renal function, we explored whether PCs critically depend on other organelles linked to lipid metabolism. While medium and long-chain fatty acids (FAs) can be metabolised directly by mitochondria, very-long-chain FAs (VLCFAs, >C22) must be broken down almost exclusively by peroxisomal β-oxidation into shorter chain FAs, before shuttling to mitochondria (Fig. 3a)[39,40]. Peroxisomes consist of a single membrane bilayer surrounding a dense enzyme matrix and a crystalline Urate Oxidase core[41]. Intriguingly, cells of the mammalian kidney contain the highest density of peroxisomes in the human body[38], however, the functional relevance of peroxisomes in the mammalian kidney remains somewhat unclear due to the challenges associated with targeted mutagenesis and metabolic plasticity[42].

Here, we exploited the genetically-encoded SKL-GFP (SKL, a peroxisomal targeting signal) reporter to visualise peroxisomes live, at subcellular resolution, in PCs and SCs within the main segment of intact renal tubules (Fig. 3b–d). Renal tubules possess vast numbers of peroxisomes (Fig. 3b), which are found at significantly higher levels in main segment PCs than SCs (Fig. 3c–e). High resolution time-lapse imaging revealed renal peroxisomes displayed highly dynamic behaviour (Fig. 3f and Movie S1) and regularly extended dynamic, pexopodial-like projections as they interacted with adjacent lipid droplets (Fig. 3f–h and Movie S2), a phenomenon that may facilitate exchange and metabolism of cellular lipids[43,44]. Indeed, PC peroxisomes were found in close proximity to key metabolic compartments, the lipid droplets and mitochondria (Fig. 3i), which were all more enriched within PCs than SCs. Consistent with PC peroxisomal enrichment, PCs expressed significantly higher levels of key peroxisome biogenesis genes than their SC counterparts (Fig. 3j). Whilst it is known from in vitro studies that cellular peroxisome volume (or number) changes depending on nutritional conditions[45], the exact role of peroxisomes' physical and dynamic interactions with other organelles in vivo is poorly understood. Although a more detailed understanding of peroxisomal dynamics exists in plant models[46], until now, most studies on peroxisome movement in animal (mammalian and invertebrate) systems have utilised isolated cells in vitro[47].

The enrichment of peroxisomes within the PCs suggest that they may be further supplementing their FA supply through peroxisomal β-oxidation. Indeed, PCs express significantly higher levels of genes

encoding rate-limiting peroxisomal FAO enzymes that catalyse key steps of VLCFA β-oxidation than their SC neighbours (such as acyl-coA oxidase, Acox3; Fig. 3j, Fly Cell Atlas single nucleus transcriptomic data). To explore the role of renal peroxisomes and their role in β-oxidation, we inhibited peroxisomal biogenesis specifically in PCs through RNAi-mediated inhibition of conserved Pex genes that play key roles in peroxisome proliferation[48] (Fig. 3k–m). RNAi-mediated inhibition of *Pex3* or *Pex16* in PCs significantly perturbed tubule secretory activity and PC bioenergetic output (Fig. 3k, l). This peroxisome disturbance was associated with a corresponding increase in cellular lipid droplet content (Fig. 3m–o), suggesting there is an intimate, often reciprocal, relationship between peroxisomes and cellular lipid stores; since excessive FAs can induce mitochondrial uncoupling and dysfunction, free FAs are often sequestered as triacyl-glycerol (TAG) in intracellular lipid droplets (LDs) to protect against lipotoxicity[37]. Excess FAs have also been shown to trigger peroxisomal proliferation, possibly via the lipid sensing hormone receptor PPAR family[49].

As well as being a major site of FAO, peroxisomes traditionally act as storage for key cellular enzymes involved in redox balance and waste detoxification[50] (Fig. 3a). In *Drosophila* renal tubules, the peroxisomal enzymes Xanthine dehydrogenase (XDH), Xanthine oxidase and Urate oxidase (Uro) play essential roles in purine catabolism (generating the waste product uric acid) and xenobiotic metabolism (Mortia 1958); in fact, mutation of XDH or Uro causes dramatic renal defects, with the accumulation of xanthine stones and uric acid nephrolithiasis[51]. As a consequence of FAO and oxidase activity, peroxisomes also generate significant ROS[52], but to counter this, peroxisomes possess vast concentrations of antioxidant enzymes, such as Catalase, Glutathione peroxidase and Superoxide dismutase[53]. Consistent with this, renal PCs are significantly more enriched in these key peroxisomal enzymes, both those involved in waste detoxification and redox balance, compared to their SC neighbours (Fig. 3j). The striking decline in renal function following peroxisomal disturbance in PCs (Fig. 3k) may thus reflect their key roles in both energy production and waste detoxification. Intriguingly, the peroxisomes at the distal end of the renal tubule (in the so-called 'initial' segment) appear to have been co-opted for a different role in calcium storage excretion[54].

## Pentose phosphate pathway supports PC antioxidant priming and limits premature ageing

Whilst PCs can import significantly more glucose than their SC neighbours, our data suggest imported glucose is not essential for mitochondrial ATP production, implying that PCs may divert glucose (or its derivative glucose-6-phosphate, G6P) into alternative metabolic pathways. The PPP (Fig. 4a) facilitates reductive biochemistry within the cell via NADPH regeneration[55]; NADPH is a crucial electron donor required for reductive biosynthesis, which includes the regeneration of antioxidant compounds such as reduced glutathione (GSH), the major antioxidant thiol (Fig. 4a). In fact, PCs exhibit significantly increased expression of key enzymes in the PPP (Fig. 4b, Fly Cell Atlas single nucleus transcriptomic data) and contain much higher levels of both NADPH (Fig. 4c) and GSH (Fig. 4d) than adjacent SCs, suggesting that PCs might be using considerable glucose to support PPP-mediated NADPH and GSH regeneration. Indeed, renal PCs are particularly vulnerable to oxidative stress (more so than their SC neighbours) due to their unavoidable mitochondrial ROS (mtROS) production[28] (Fig. 4e); PCs show moderate signs of oxidative stress, such as lipid peroxidation, even in homeostasis (Fig. 4f), along with transcriptional activation of antioxidant response pathways (e.g., downstream of the transcription factor Nrf2, Fig. 4g).

To address whether glucose stores are being diverted towards the PPP within renal PCs, we firstly inhibited the glycolytic enzyme HexA using 2-deoxy-glucose (2DG); 2DG, a classical glycolytic inhibitor, competitively inhibits HexA[56] and leads to reduced glucose-6-P

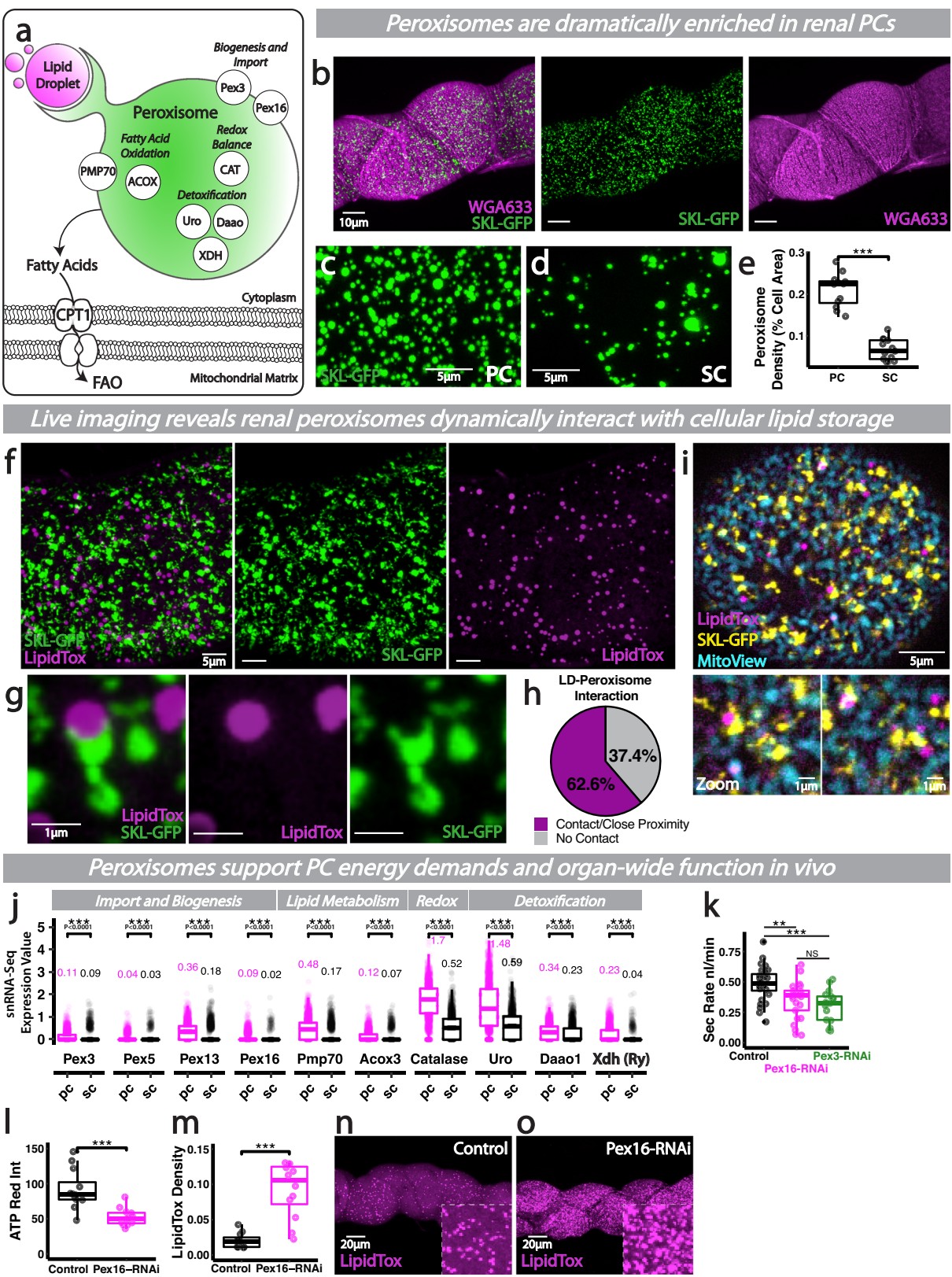

production, which in turn limits flow into the PPP (Fig. 4a). Intact, living renal tubules treated with 2DG exhibited markedly reduced excretory activity (Fig. 4h) that was associated with reduced GSH regeneration (Fig. 4i) but no change in ATP production (Fig. 4j) within constituent PCs. Although 2DG inhibits G6P flux into both glycolysis and PPP, our data indicate that PGI (and thus glycolysis downstream of PGI) is not required for tubule function (Fig. 1t, u), suggesting that PC glucose is

being directed towards PPP to sustain tubule function. Given renal vulnerability to oxidative stress, we envision that PCs may programme their metabolism in this way to support regeneration of key anti-oxidant factors.

To further investigate a role for the PPP in renal PCs, we inhibited PPP specifically in PCs using RNAi-mediated knock-down of the *Droso-phila* G6P dehydrogenase (dG6PD; also known as *zwishenferment, zw*)

**Fig. 3 | Dynamic lipid-peroxisomal networks support PC ATP synthesis and renal function. a** Schematic of key peroxisome proteins and functions. **b**–**e** Peroxisomes (SKL-GFP) and analysis and quantification of peroxisome density in PCs and SCs. **f**–**h** Peroxisomes (SKL-GFP) interacting with lipid droplets (Lipid-Tox) in PCs via putative pexapodia, along with quantification. **i** Peroxisomes (SKL-GFP), lipid droplets (LipidTox) and mitochondria (MitoView) in PCs, high magnification examples of tight inter-organelle associations. **j** Expression of key peroxisome proteins in PCs and SCs from publicly available snRNA-Seq data (Fly Cell Atlas). Secretory activity (**k**), bioenergetic output (**l**) and LD density (**m**–**o**) in *Pex3* or *Pex16* RNAi tubules (driven by *CapaR-gal4*). Data represented as box and whisker plots (lower and upper hinges correspond to the first and third quartiles, median line within the box, whiskers extend from the hinge to the largest/smallest value, at most 1.5* interquartile range of the hinge) with all data from MpT cells (SCs or PCs, **e**), MpT main segment sections (**l**, **m**) or secretion of individual kidneys (**k**) shown as overlaid points. NS Not Significant, **$P < 0.01$, ***$P < 0.001$ (unpaired two-tailed t-tests, ANOVA followed by Tukey multiple comparisons test where appropriate or Wilcoxon test with FDR correction). p values: (**e**) $p < 0.0001$, (**k**) control vs Pex16 $p = 0.00288$, control vs Pex3 $p = 0.000303$, (**l**) $p = 0.000484$, (**m**) $p < 0.0001$. p values for (**j**) displayed on the Figure. For analysis of fluorescent reporters/dyes, two images of different sections of the MpT main segment per fly were imaged. All images representative of >5 tubules. All images are maximum z projections. **e** $n = 12$ PCs (from 6 tubules) and 13 SCs (from 5 tubules), (**h**), $n = 7$ tubules, (**k**), $n = 28$ control, 25 Pex16 and 19 Pex3 tubules, (**l**) $n = 12$ tubules per condition, (**m**) $n = 11$ control and 12 RNAi tubules. Source data are provided as a Source Data file.

or *Drosophila* PGD (Fig. 5a). Inhibition of PPP within PCs caused a significant reduction in each cell's ability to generate GSH (Fig. 5b–d) and this was associated with a dramatic decline in renal-wide secretory activity (Fig. 5e). However, inhibition of PPP in SCs (C724>dG6PD-RNAi) had no significant impact on tubule secretory activity (Fig. S3a). We envision that GSH depletion in PCs may be a major contributor to tubule functional decline. Mitochondria also rely on GSH to maintain a homeostatic redox state but lack the ability to synthesise it and must rely on import from the cytosol[57]; starving GSH regeneration (via PPP inhibition) may therefore limit the GSH pool available for mitochondrial import, creating an unfavourable mitochondrial environment (Fig. S3b, JC-1 levels suggesting reduced mitochondrial activity).

Renal tubules with PPP-deficient PCs exhibited prominent morphological defects (Figs. 5f, g and S3c-d), with dramatically enlarged PC nuclei (Figs. 5f, g, 5j and Fig. S3c-e) and increased cell area (Figs. 5k, S3f) as well as disturbed apical (luminal) membranes (Fig. S3g, h). Cellular swelling and enlarged nuclei are often suggestive of cellular senescence in flies and mammals[58]. PPP-deficient PCs also exhibited several other phenotypes characteristic of a senescence-like state, including elevated nuclear H3K9me3 (a repressive epigenetic mark, Fig. 5l–n), increased lysosomal density (Fig. 5o–q) and enhanced SA-βGal activity (Fig. 5r–t and Fig. S3i–k). Crucially, these PPP-deficient phenotypes appear to be degenerative, appearing progressively during adult tubule function (rather than appearing during development), as they were not observed in renal tubules from freshly hatched (1-day old) adults (Fig. 5h–k). This suggests that PPP activity is most crucial in PCs in adult tubules when the tubules are energetically demanding and physiologically active[28].

## Differential estrogen-related receptor activity may couple cellular identity to metabolism within complex heterogenous tissue in vivo

We next explored whether the striking metabolic differences between PCs and SCs were hard-wired into each cellular identity (e.g., by the action of cell-type specific TFs). When searching for potential candidate genes that were differentially expressed within PCs and SCs, we discovered that the *Drosophila* ERR homolog, dERR, was particularly enriched within the renal tubules ('MpT Cell', Fig. 6a) compared to other non-renal tissues (using Fly Cell Atlas transcriptomic data of adult *Drosophila*[26]). Moreover, dERR expression was significantly higher within PCs compared to SCs, both at the RNA (Fig. 6b, using Fly Cell Atlas single nucleus transcriptomic data[19]) and protein level (Fig. 6c–e, using the dERR-GFP reporter where GFP is fused to the endogenous ERR protein). Intriguingly, ERR family transcriptional regulators are known to play key roles directing organism- or tissue-wide developmental switches in mammals and *Drosophila*[59–61], where they couple developmental cellular differentiation and energy production by co-regulating the expression of genes for cellular function and mitochondrial biogenesis[62,63]. We thus explored whether differential ERR expression between cells within a single heterogenous tissue can establish (and sustain) distinct cellular identities and metabolic profiles.

To explore a role for ERR, we knocked-down ERR using RNAi specifically in PCs (the cell type in which it was enriched) and this caused a marked reduction in tubule secretory activity (Fig. 6f). To investigate whether ERR supported tubule secretory function via effects on PC metabolism, we explored glucose and lipid handling in ERR-deficient PCs. Live-imaging of ERR-deficient PCs revealed a much reduced capacity for glucose uptake (Fig. 6g–i) and a loss of lipid droplet stores (Fig. 6j–l). In contrast, ERR-deficient PCs were far more efficient at taking up extracellular FAs (as shown via BODIPY C12, Fig. 6m–o), suggesting that while ERR loss may reduce PC's capacity to store lipids in LDs (Fig. 6j–l) it may also drive compensatory FA uptake. Indeed, in the absence of ERR, BODIPY C12 is localised in a more diffuse, cytosolic pattern rather than in LD-like punctae, suggesting that ERR may play a crucial role in programming storage of the PC lipid pool. ERR thus appeared to regulate multiple key aspects of PC metabolism. Consistent with the observed overall reduction in tubule excretory activity, ERR-deficient PCs displayed a marked elevation in ATP levels (Fig. 6p–r) which perhaps reflects lack of V-ATPase activity. Intriguingly work in yeast has shown V-ATPase assembly is proportional to intracellular glucose concentration and the flux of glycolysis[64], raising the possibility that the lack of intracellular glucose in ERR-deficient PCs might facilitate V-ATPase disassembly, leading to reduced ATP hydrolysis and contributing to defective tubule secretion. Increased ATP production has also been proposed as a dramatic 'hyper-metabolism'-like response in stressed cells with dysregulated metabolism (e.g., OXPHOS defects)[65].

During early development, dERR has been linked to direct regulation of genes encoding glycolytic, lipogenic and PPP components[59]. In mature renal tubules, cell type-specific ERR signalling could ensure coordinated activation of genes involved in metabolism with cell-specific energy-consuming processes (e.g., active transport), to ensure renal cell capacity for energy production is precisely matched to functional energy demands (Fig. 6s). Indeed, identification of ERRα targets in the mammalian kidney revealed that the Transcription factor Cut1-like is an ERRα target gene[66]. The *Drosophila* homeobox transcription factor Cut has well-known roles in specifying cellular identity during renal development and is highly enriched within PCs compared to SCs[28,67]. *Drosophila* Cut was recently shown to support region-specific mitochondrial network configuration in muscles[68]. Perhaps ERR, together with Cut, couples PC cell fate (via expression of specific transporters or ion channels) with precisely tuned metabolism.

## Metabolic dysfunction and ERR loss accompany progressive age-related renal decline in vivo

With an ageing population, increasing numbers of individuals are being diagnosed with age-related kidney impairment; these individuals also have a higher susceptibility to AKI and increased likelihood to progress to CKD[69]. However, deciphering the progressive metabolic changes that occur in the ageing mammalian kidney is technically challenging, with the majority of studies to date performed on healthy kidneys, artificially-aged kidneys or those with CKD. CKD is itself linked to defects in metabolism, particularly metabolic inflexibility[70], with

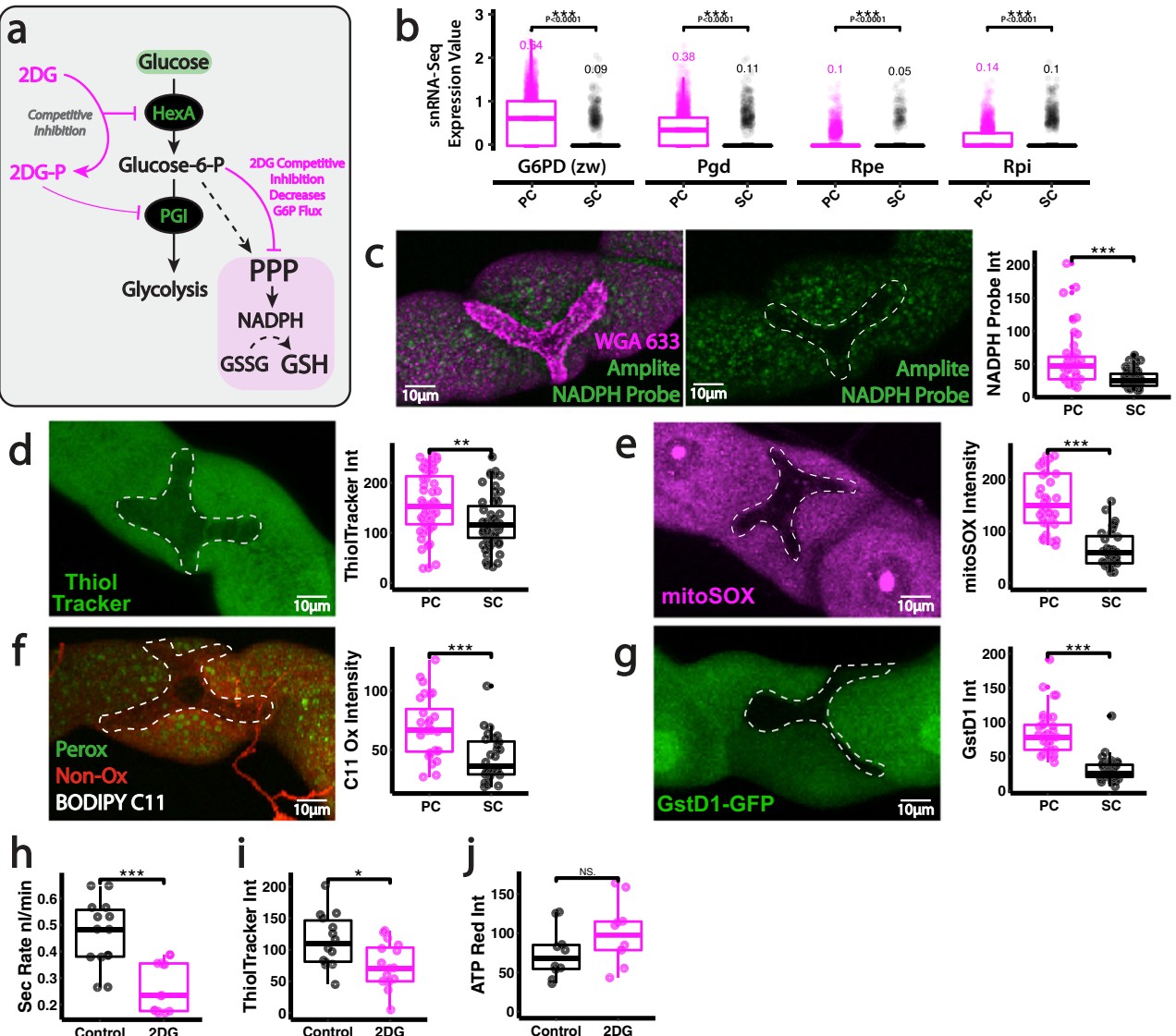

**Fig. 4 | Sustained PPP activity within renal PCs supports organ-wide function.**
**a** Schematic of early glucose metabolism and mechanism of inhibition.
**b** Expression of key PPP components in PCs and SCs from publicly available Fly Cell Atlas snRNA-Seq data. Analysis and quantification of Amplite NADPH probe staining (**c**), Thioltracker (GSH) (**d**), mitoSox (**e**), BODIPY C11 (**f**) and GstD1-GFP (**g**) in renal tubule SCs and PCs. Secretion rate (**h**), bioenergetic output (ATP Red, **i**) and Thioltracker (GSH, **j**) staining in 2DG treated renal tubules. Data represented as box and whisker plots (lower and upper hinges correspond to the first and third quartiles, median line within the box, whiskers extend from the hinge to the largest/smallest value, at most 1.5* interquartile range of the hinge) with all data from MpT cells (SCs or PCs, **b**–**g**), MpT main segment sections (**i**, **j**) or secretion of individual kidneys (**g**) shown as overlaid points. NS Not Significant, *$P < 0.05$,

**$P < 0.01$, ***$P < 0.001$ (unpaired two-tailed $t$-tests or Wilcoxon test with FDR correction). $p$ values: (**c**) $p < 0.0001$, (**d**) $p = 0.00258$, (**e**) $p < 0.0001$, (**f**) $p = 0.000117$, (**g**) $p < 0.0001$, (**h**) $p < 0.0001$, (**i**) $p = 0.0119$. $p$ values for (**b**) displayed on the Figure. For analysis of fluorescent reporters/dyes, two images of different sections of the MpT main segment per fly were imaged. All images representative of >9 tubules. All images are maximum z projections. **c** $n = 46$ cells (from 11 tubules) per condition, (**d**) $n = 55$ cells (from 7 tubules) per condition, (**e**) $n = 36$ cells (from 6 tubules) per condition, (**f**) $n = 28$ cells (from 7 tubules) per condition, (**g**) $n = 35/36$ cells (from 6 tubules) per condition, (**h**) $n = 14$ tubules per condition, (**I**) $n = 14$ control and 16 2DG tubules, (**j**) $n = 10$ tubules per condition. Source data are provided as a Source Data file.

transcriptomics of patient samples suggesting striking dysregulation of genes involved in lipid metabolism, FAO and OXPHOS[25,71,72]. Recent profiling of donor kidneys has identified metabolism-related proteomic signatures associated with age that correlate with sub-optimal post-transplant function[23]. The major challenge is now identifying which of these age-related changes play causal roles in renal decline in vivo.

Given our data suggesting differential ERR activity directs renal metabolic programming, age-related changes in ERR could, at least in part, drive renal dysfunction. We thus examined how renal ERR expression and cellular metabolism is progressively altered during natural ageing in vivo. Renal tubules displayed a significant reduction

in ERR protein levels (Fig. 7a, mass spectrometry data from entire intact tubules) and a progressive decline in excretory activity (Fig. 7b) with increasing age. These functional changes were accompanied by striking alterations in multiple metabolic networks, including those related to lipid metabolism, glucose handling and the PPP, which mimic the defects we observed following experimental inhibition of tubule ERR and PPP activity. Individual PCs within intact living aged (28-day) renal tubules exhibited markedly reduced ability to uptake glucose (Fig. 7c–e) and also exhibited signs of dyslipidaemia. Whilst aged PCs had a higher capacity to import cellular lipids (Fig. 7f–h), they exhibited significantly reduced total cellular lipid stores (lipid droplets, LDs) compared to their younger counterparts (Fig. 7i–k). This

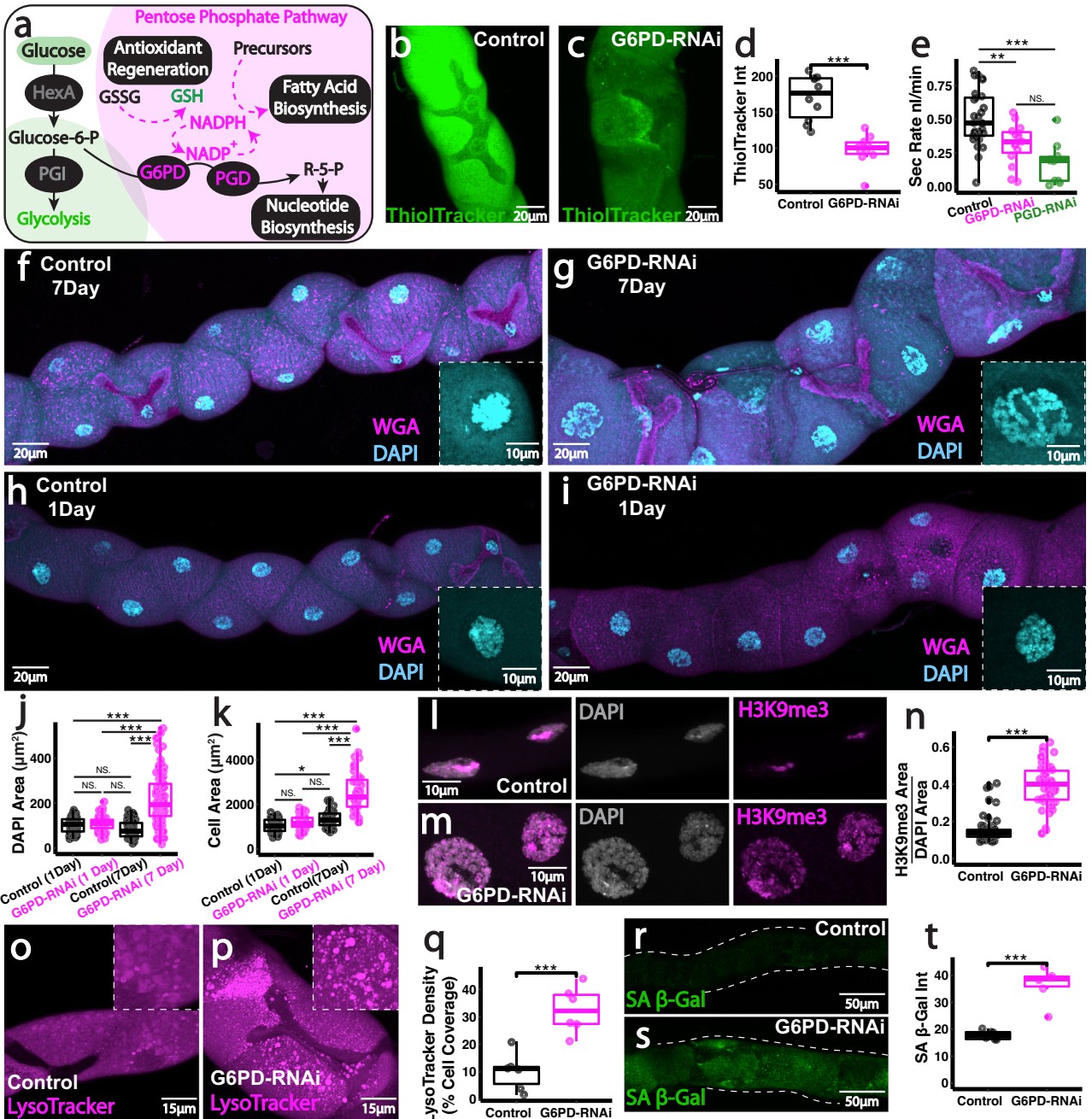

**Fig. 5 | PPP activity primes PC antioxidant defences and limits premature renal aging. a** Schematic of the key roles of the pentose phosphate pathway. Analysis and quantification of Thioltracker (GSH, **b**–**d**) and secretory activity (**e**) in *G6PD-RNAi* and *PDG-RNAi tubules* (driven by *CapaR-Gal4*). **f**, **g** WGA and DAPI staining in *G6PD-RNAi* tubules at 7 days and 1 day (**h**, **i**) post-eclosure. **j**, **k** Quantification of nuclear area (DAPI) and cell area. **l**–**n** H3K9me3 and DAPI, Lysotracker **o**–**q** and SA β-Gal (**r**–**t**) staining and quantification in 7-day *G6PD-RNAi* tubules. Data represented as box and whisker plots (lower and upper hinges correspond to the first and third quartiles, median line within the box, whiskers extend from the hinge to the largest/smallest value, at most 1.5* interquartile range of the hinge) with all data from MpT main segment sections (**d**, **q**, **t**), nuclei/cells from main segments (**j**, **k**, **n**) or secretion of individual kidneys (**e**) shown as overlaid points. NS Not Significant, **P < 0.01, ***P < 0.001 (unpaired two-tailed *t*-tests or ANOVA followed by Tukey multiple comparisons test where appropriate). *p* values: (**d**) *p* < 0.0001, (**e**) control vs G6PD *p* = 0.00507, control vs PGD *p* < 0.0001, (**j**) 1 day

control vs 7 day G6PD *p* < 0.0001, 1 day G6PD vs 7 day G6PD *p* < 0.0001, 7 day control vs 7 day G6PD *p* < 0.0001, (**k**) 1 day control vs 7 day control *p* = 0.00514, 1 day control vs 7 day G6PD *p* < 0.0001, 7 day G6PD vs 1 day G6PD *p* < 0.0001, 7 day G6PD vs 7 day control *p* < 0.0001, (**n**) *p* < 0.0001, (**q**) *p* = 0.000567, (**t**) *p* = 0.000574. *p* values where *p* > 0.05 labelled as NS. For analysis of fluorescent reporters/dyes, two images of different sections of the MpT main segment per fly were imaged. All images representative of >5 tubules. All images are maximum z projections. **d** *n* = 10 tubules per condition, (**e**) *n* = 29 control, 15 G6PD and 10 PGD tubules, (**j**) 73 nuclei (from 6 control 1 day tubules), 66 nuclei (from 4 G6PD 1 day tubules), 68 nuclei (from 5 control 7 day tubules) and 97 nuclei (from 9 G6PD 7 day tubules), (**k**) 70 cells (from 6 control 1 day tubules), 61 cells (from 4 G6PD 1 day tubules), 50 cells (from 5 control 7 day tubules) and 45 cells (from 9 G6PD 7 day tubules), (**n**) *n* = 56/50 nuclei (from 4 tubules) per condition, (**q**) *n* = 6 tubules per condition, (**t**) *n* = 6 tubules per condition. Source data are provided as a Source Data file.

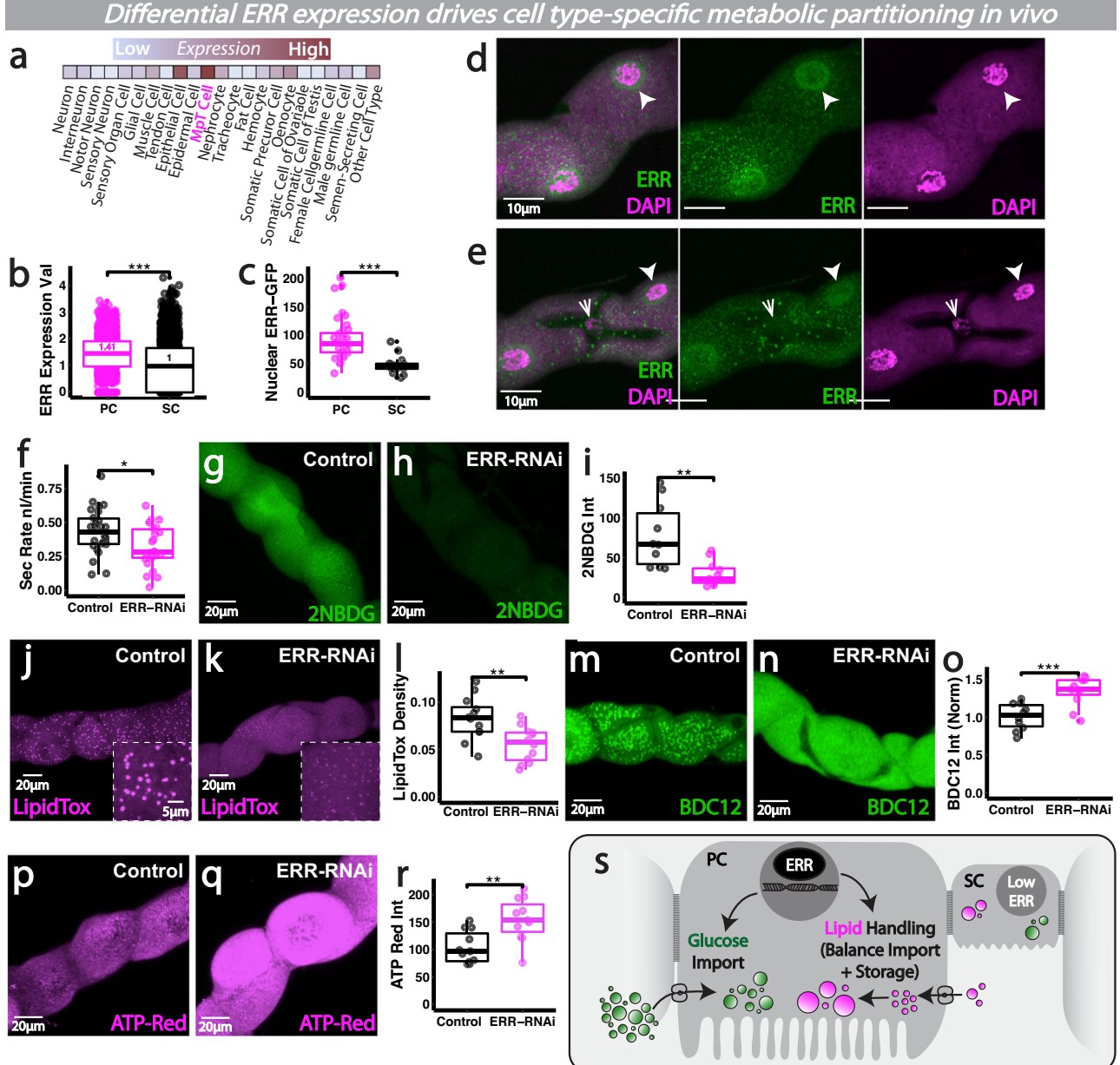

**Fig. 6 | Differential ERR activity may couple cellular identity to metabolism within intact heterogenous renal tissues. a** Fly Cell Atlas scRNA-seq expression of ERR. **b** Cell-type specific snRNA-seq expression of ERR from publicly available Fly Cell Atlas snRNA-seq data. **c–e** *ERR-GFP* and DAPI analysis and quantification in renal SC and PC nuclei. **f** Secretory activity in *CapaR>ERR-RNAi* tubules. **g–i** Analysis and quantification of 2-NBDG, Lipidtox (**j–l**), BODIPY C12 (**m–o**) and ATP-Red (**p–r**) in *ERR-RNAi* tubules. **s** Schematic of ERR metabolic programming. Data represented as box and whisker plots (lower and upper hinges correspond to the first and third quartiles, median line within the box, whiskers extend from the hinge to the largest/smallest value, at most 1.5* interquartile range of the hinge) with all data from MpT main segment sections (**i**, **l**, **o**, **r**), nuclei from main segments (**c**) or secretion of

individual kidneys (**f**) shown as overlaid points. NS Not Significant, *$P < 0.05$, **$P < 0.01$, ***$P < 0.001$ (unpaired two-tailed *t*-tests or Wilcoxon test with FDR correction). *p* values: (**b**) $p < 0.0001$, (**c**) $p < 0.0001$, (**f**) $p = 0.0171$, (**i**) $p = 0.00446$, (**l**) $p = 0.00358$, (**o**) $p = 0.000584$, (**r**) $p = 0.00285$. For analysis of fluorescent reporters/dyes, two images of different sections of the MpT main segment per fly were imaged. All images representative of >9 tubules. All images are maximum z projections. **c** $n = 36/14$ nuclei (from 5 tubules), **f** $n = 26$ control and 27 RNAi tubules, **i** $n = 10$ tubules per condition, **l** $n = 12$ tubules per condition, **o** $n = 10/11$ tubules per condition, (**r**) $n = 11$ tubules per condition. Source data are provided as a Source Data file.

loss of cellular lipid storage was accompanied by a marked age-related rise in the peroxisomal content of PCs (Fig. 7l–n). To date, a large proportion of molecular studies on aging have focused on the role of mitochondria and their dysfunction, whilst neglecting the contribution of other major metabolic organelles, such as the peroxisomes[73]. Given *Drosophila* with reduced peroxisomal content had increased lifespan[74], the dramatic expansion of peroxisomal content in aged renal tubules could be detrimental in vivo. Indeed, the activity of

peroxisomal ACOX1, the rate-limiting step in peroxisomal β-oxidation, results in extensive $H_2O_2$ generation which may, if unrestrained, drive debilitating cellular damage[75].

PCs within aged tubules also exhibited much reduced levels of the antioxidant GSH compared to those in young, 7-day tubules (Fig. 7o–q), suggestive of reduced flux through the PPP. In parallel, PCs within aged renal tubules accumulated significantly higher levels of mitochondrial ROS (Fig. 7r–t), perhaps as a consequence of the

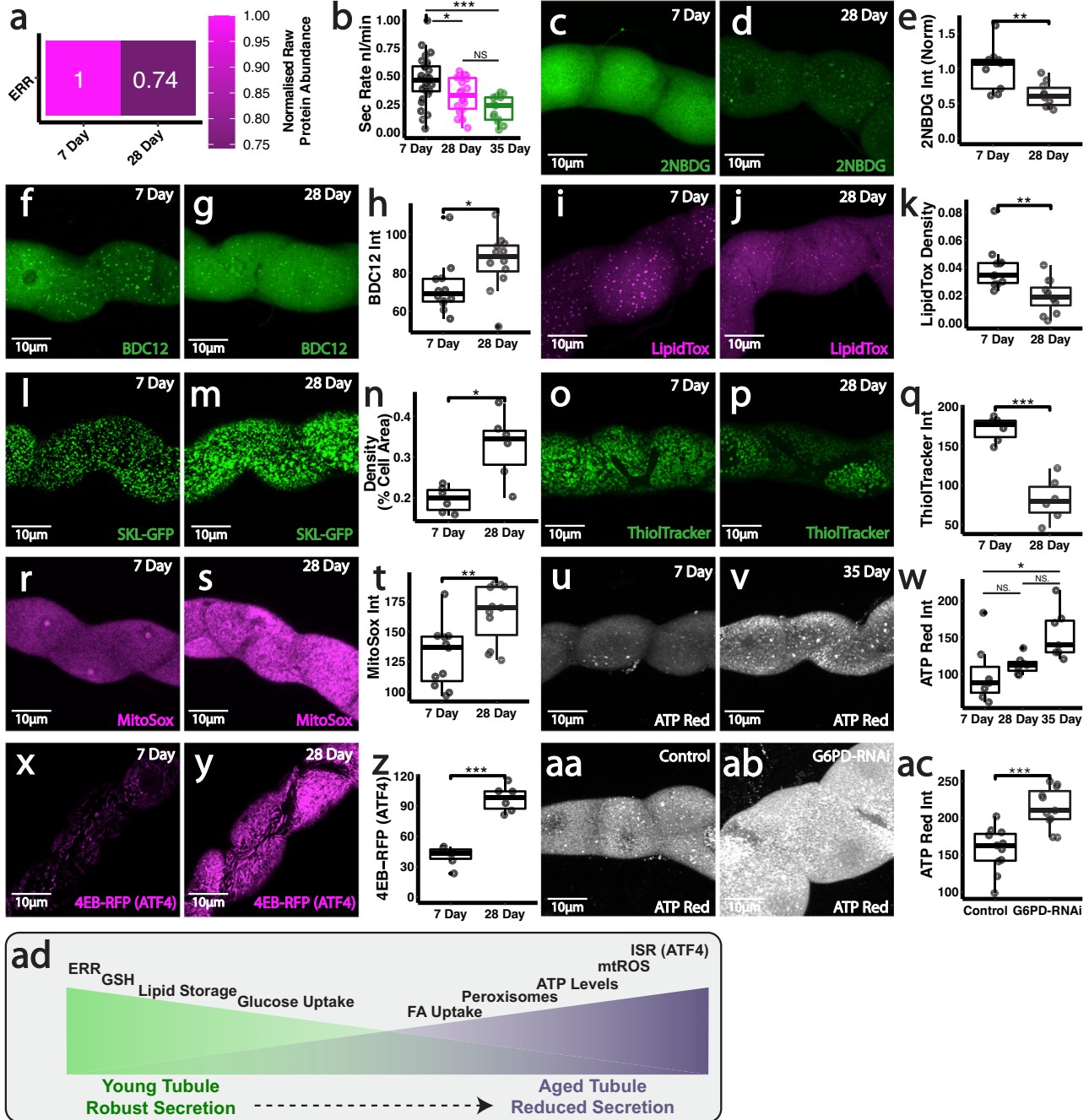

**Fig. 7 | Metabolic dysfunction and reduced ERR activity accompany progressive age-related renal decline in vivo. a** Relative ERR protein abundance in 7 vs 28 day MpT tissue. **b** Analysis and quantification of secretion rate in 7, 28 and 35 day MpTs. Analysis and quantification of 2-NBDG uptake (**c**–**e**), BODIPY C12 (**f**–**h**), LipidTox density (**i**–**k**), peroxisome (*CapaR>SKL-GFP*) density (**l**–**n**), Thiol-Tracker (**o**–**q**), MitoSox (**r**–**t**), ATP red (**u**–**w**) and ATF4 activity (4E-BP-dsRed, reporter of the Integrated Stress Response) (**x**–**z**). Analysis and quantification of ATP red staining in *CapaR>G6PD-RNAi* tubules (**aa**–**ac**). Summary schematic of metabolic changes associated with aging (**ad**). Data represented as box and whisker plots (lower and upper hinges correspond to the first and third quartiles, median line within the box, whiskers extend from the hinge to the largest/smallest value, at most 1.5* interquartile range of the hinge) with all data from MpT main segment sections (**e**, **h**, **k**, **n**, **q**, **t**, **w**, **z**, **ac**), or secretion of individual kidneys (**b**) shown as overlaid points. NS Not Significant, *$P < 0.05$, **$P < 0.01$, ***$P < 0.001$ (unpaired two-tailed $t$ tests or ANOVA followed by Tukey multiple comparisons test where appropriate). $p$ values: (**b**) 7 vs 28 day $p = 0.0232$, 7 vs 35 day $p = 0.000168$, (**e**) $p = 0.00759$, (**h**) $p = 0.0282$, (**k**) $p = 0.00258$, (**n**) $p = 0.0101$, (**q**) $p = 0.000123$, (**t**) $p = 0.00369$, (**w**) 7 vs 35 day $p = 0.0148$, (**z**) $p < 0.0001$, (**ac**) $p = 0.000149$. $p$ values where $p > 0.05$ labelled as NS. For analysis of fluorescent reporters/dyes, two images of different sections of the MpT main segment per fly were imaged. All images representative of >5 tubules. All images are maximum z projections. **b** $n = 29$ 7 day, 22 28 day and 13 35 day tubules, (**e**) $n = 11/12$ tubules per condition, (**h**) $n = 12$ tubules per condition, (**k**) $n = 12/13$ tubules per condition, (**n**) $n = 6$ tubules per condition, (**q**) $n = 6$ tubules per condition, (**t**) $n = 11$ tubules per condition, (**w**) $n = 7$ tubules per condition, (**z**) $n = 6$ tubules per condition, (**ac**) $n = 11$ tubules per condition. Source data are provided as a Source Data file.

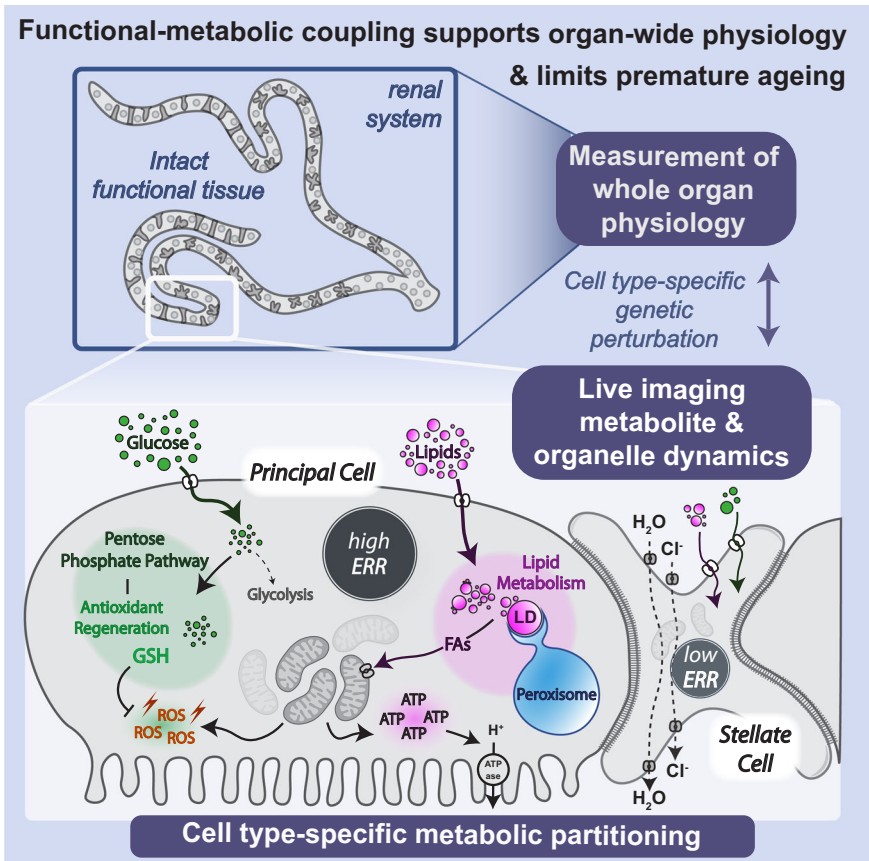

**Fig. 8 | Summary schematic.** Schematic summarising how mechanistic analysis of single cell metabolism can be linked to readouts of whole organ physiological function, using the *Drosophila* renal tubules. We propose the renal tubules as a paradigm tissue for exploring functional metabolic coupling and partitioning within an intact living tissue.

reduced antioxidant capacity of these aged cells. Despite the overall renal functional decline, ATP levels within PCs surprisingly increased with age (Fig. 7u–w), similar to that observed following *ERR-RNAi*. This was accompanied by a dramatic upregulation in the activity of ATF4 (Fig. 7x–z), a stress responsive transcription factor implicated in the integrated stress response (ISR)[76] and recently suggested to drive cellular injury in human renal cells[77]. Intriguingly, age-associated metabolic inefficiency has recently been shown to trigger the ISR and cause 'hyper-metabolism' due to the enormous energetic cost of mounting these wide-scale stress responses[65]. By reducing energy available for other cellular processes, ISR and hyper-metabolism have been linked to rapid age-related physiological decline and cellular senescence[65,78].

These metabolic changes observed during natural renal ageing, are remarkably similar to those we observed following experimental inhibition of ERR and PPP (Figs. 4–6); indeed, tubules lacking a functional PPP appeared to undergo a dramatic acceleration of renal aging and induction of senescence (Fig. 5). Intriguingly, PPP-deficient PCs also exhibited a surprising elevation in ATP production (Fig. 7aa–ac), suggesting potential onset of a hyper-metabolic state, despite a decline in secretory activity, as in ERR deficient tubules. It remains unclear whether a detectable increase in cellular biomarkers typically associated with cellular senescence also occurs in naturally aged 28-day or 35-day renal tubules; we speculate that metabolic dysfunction and tubule functional decline may precede the onset of renal senescence in vivo. Ultimately, this work identifies ERRs as important modulators of cell-type specific metabolism in the kidney essential to support organ-wide function, whose decline over the life-course may underlie age-related renal dysfunction (Fig. 7ad).

## Discussion

Despite the long-held belief that complex heterogenous tissues (like the kidney) possess enormous diversity not only in cellular function but also in metabolic profiles, defining these distinct metabolic programmes at the level of individual cells within their native in vivo tissue context - and testing their relevance to organ function - has been limited by the complexity and experimental challenges associated with traditional mammalian models. A comprehensive understanding of the role of cell-type specific metabolism in complex living biological systems is therefore still elusive. Here, the comparatively simple renal tubules of *Drosophila* offer unrivalled opportunities to interrogate how spatially coupled, yet functionally distinct, cell types fine-tune their metabolic profiles in vivo to support their unique functional roles and coordinate tissue-wide physiology (Fig. 8).

By integrating cell-type specific genetic perturbation, coupled with high resolution live-imaging and physiological assays within the intact living tubule, we reveal that strict functional-metabolic coupling within distinct renal cell types supports renal function and limits premature ageing. Since kidneys are amongst the most energetically-demanding body tissues and renal function is paramount to organism survival, we anticipate that functional-metabolic coupling is of particularly prime importance in this tissue. We find precise programming not only ensures sufficient ATP is generated *efficiently* via FAO by cells with the greatest metabolic demand (the active transport-dependent PCs), but that potentially toxic oxidative metabolism is balanced by simultaneous activation of robust antioxidant measures (e.g., via GSH regeneration via the PPP) to limit premature senescence. Moreover, hard-wiring of this metabolic programming appears to be reliant on the transcription factor ERR,

as RNAi-mediated ERR knockdown disrupts these differentially partitioned metabolic states.

It is particularly striking that metabolic adaptations in one cell type (the PCs) play such pivotal, non-autonomous roles in supporting organ-wide physiology; indeed, renal fluid secretion is severely inhibited following disruption of multiple aspects of PC metabolism, despite water transport almost exclusively occurring through SCs[18]. Undoubtedly this reflects the central role of PCs in creating effective ion gradients to drive SC-mediated water flow. Strategic separation of metabolically-intense cation transport in PCs and water transport via aquaporins in SCs[18] may also protect surrounding tissues from collateral damage. Indeed, as well as conducting water movement, aquaporins are increasingly linked to mediating passive movement of ROS (such as hydrogen peroxide)[79]. By preferentially restricting aquaporins to SCs which exhibit limited metabolic activity (and thus minimal mitoROS production), this may curb the potential for dangerous ROS leakage into surroundings.

The rapid rerouting of glucose from glycolysis to the oxidative arm of the PPP is an emerging adaptive response of mammalian cells to oxidative stress, triggered in part by the blockade of lower glycolysis by direct oxidation of key enzymes (e.g., glyceraldehyde 3-phosphate dehydrogenase)[80]. Our data suggest that rather than being a temporary adaptive stress response, renal PCs may constitutively inhibit lower glycolytic enzymes and divert glucose into the PPP. Intriguingly, the PPP intermediate E4P is a known inhibitor of glycolytic PGI, suggesting cells with an active PPP may feedback to restrain glycolysis and ensure continued glucose flux into the PPP. As well as playing a crucial role in regenerating cellular GSH, NADPH promotes production of precursor metabolites (such as ribose 5-phosphate and erythrose 4-phosphate) for the synthesis of nucleotides, aromatic amino acids and histidine[55]. Given that PCs are well-known to undergo endoduplication to achieve polyploidy[81], PPP activity in these cells may also support nucleotide synthesis. The intersection of the PPP with many biosynthetic processes, might therefore closely link not only antioxidant regeneration but also energy metabolism and genome maintenance in these cells, so PPP is not surprisingly key to maintaining renal function and prolonging healthy ageing.

Our ex vivo live-imaging revealed that renal PCs are densely packed with peroxisomes, lipid droplets and mitochondria that display striking dynamics (including inter-organelle interactions), opening up *Drosophila* as a powerful system in which to study the physical and functional relationships between these organelles. Lipid droplets enriched in PCs may serve additional important protective functions to mitigate oxidative damage, both within the tubule and more systemically. Indeed, lipid droplets within *Drosophila* podocyte-like cells (nephrocytes) afford stress protection[13] and in *Drosophila* neuroblasts, the re-localisation of polyunsaturated FAs into lipid droplets upon oxidative stress limits potentially damaging peroxidation of plasma membrane lipids[37]. *Drosophila* tubules themselves clear lipids from the circulating blood to prevent the accumulation of tissue-damaging oxidised lipids[82]. PC lipid droplets could also function as valuable energetic reserves to fuel more intense renal activity in response to environmental or microbial challenge.

Metabolic differences between renal cell types might not only support differential energy requirements, but may also be intimately linked with cellular function, as metabolic intermediates may reinforce functional differences between cells. Indeed, metabolic pathways classically only thought to play a role in energy output are emerging as central signalling hubs that shape cellular function and tissue architecture by influencing transcription, chromatin and signal transduction independently of energy production[83,84]. Not only is the trafficking and assembly of V ATPase regulated by early glycolytic intermediates[33,85], but cells lacking functional V ATPase stabilise Hypoxia-inducible factors (HIF-1a and HIF-1b) to drive metabolic reprogramming with elevated glycolysis and reduced OXPHOS[64].

Perhaps the lack of V ATPase activity within renal SCs helps direct their metabolism towards a more glycolytic state. It is intriguing that renal SCs express high levels of late-stage glycolytic enzymes, despite their comparative lack of glucose uptake. It is tempting to speculate that SCs may act as 'support cells' by shuttling the glycolytic product lactate to PCs for use in mitochondrial metabolism, similar to that observed between neurons and glia[86].

Our data suggest that cell-type specific metabolic differences within *Drosophila* renal tubules are driven by transcription factors (such as ERR) that set the core identity of each cell type during homeostasis. However, kidneys must adapt quickly in response to additional challenge, suggesting that the functional-metabolic coupling that supports basal renal activity might be further re-wired to match physiological activity. Nrf2, a well-known regulator of the antioxidant stress response, is increasingly being implicated in the reprogramming of metabolism[87]. We previously demonstrated that Nrf2 antioxidant activity is modulated within *Drosophila* renal tubules according to renal physiological activity[28], suggesting that Nrf2 could be well-placed to further fine-tune PC metabolism. Given the enormous energy consumption of mammalian renal (particularly proximal) tubule cells under physiological conditions, they are particularly susceptible to injury and age-related dysfunction[88]; indeed, the reliance of PT cells and fly PCs on oxidative mitochondrial metabolism to meet energy demands (and their limited capacity for glycolysis) make them susceptible to damage after ischemia and anoxia. We need to fully understand the effects of key molecules in metabolic reprogramming as they are prime targets for the early diagnosis and treatment of renal diseases and age-related renal decline.

A more precise understanding of spatially-resolved cellular metabolism is relevant well beyond the renal system. Recent seminal work has revealed that spatio-temporal compartmentalisation of glucose metabolism during embryonic development plays a crucial functional role in guiding the progression of mammalian gastrulation in vivo; whilst glucose shunt through the Hexosamine Biosynthetic Pathway supports pluripotency exit and EMT (epithelial-to-mesenchymal transition), the newly-formed mesoderm relies instead on glycolysis to fuel mesenchymal migration[89]. Stem cells can also adopt a distinct metabolic signature from restricted progenitors, directly influencing tissue homeostasis and regeneration[90]. Moreover, metabolic coupling and rewiring in a heterogenous tumour microenvironment can drive biosynthetic gains, heightened redox defences and support an overall more aggressive and persistent pathology[75].

Our understanding of metabolic networks within the context of intact, living tissues has remained somewhat elusive due to their vast complexity and propensity to rapidly shift upon perturbation, as well as a lack of accessible in vivo and ex vivo methods. We envision the relatively simple, intact organ model presented in the current study has enormous potential to provide in-depth mechanistic understanding of dynamic functional-metabolic coupling and intercellular metabolic relationships between distinct cell types within heterogenous tissues in vivo, and determine how their dysfunction contributes to disease and aging[90].

## Methods

### *Drosophila* stocks and husbandry

Fly stocks were maintained according to standard protocols [5]. All crosses were performed at 25 °C unless otherwise stated. The following Drosophila stocks were used: *Act5c-Gal4* (B#4414), *ACC-RNAi* (VDRC#108631), *CapaR-Gal4* (gift from Julian Dow), *C724-Gal4* (gift from Barry Denholm), *UAS-whd-RNAi* #1 (dCPT1, V#105400), *UAS-whd-RNAi* #2, (dCPT1, B#34066), *UAS-ERR-RNAi* (B#50686), *ERR-GFP* (B#38638), *UAS-Glut1-RNAi* (B#28645), *UAS-zw-RNAi* (dG6PD, B#50667), *GstD1-ARE:GFP* (reporter of Nrf2 activity, gift from Ioannis Trougakos), *OregonR*, *UAS-PGI-RNAi* (B#51804), *UAS-Pex16-RNAi* (B#57495), *UAS-Pex3RNAi* (B#50694), *UAS-PGD-RNAi* (B#65078), *UAS-*

*SKL-GFP* (B#28881, B#28881), *UAS-mCherry.mito.OMM* (B#66532), *4E-BP-dsRed* (reporter of ATF4 activity, generated by Kang et al., 2017[91], gift from Hyung Don Ryoo). Gifts from Irene Miguel-Aliaga: *LDH-GFP* (generated by Quinones et al., 2007), *PGI-GFP* (generated by Hudry et al., 2019), *Treh-GFP* (B#59825), *MalA1-GFP* (VDRC#318296), *R2R4-Gal4, UAS*-Laconic[11]. *Drosophila* mutants and transgenic lines were obtained from the Bloomington Stock Centre unless otherwise stated.

### Tubule secretion ('Ramsay') assays
Fluid secretion (Ramsay) assays were performed using live MpTs dissected from flies (of appropriate age) in ice-cold Schneider's medium (Sigma-Aldrich, S0146 with L-glutamine and sodium bicarbonate)[28]. MpTs were transferred to Schneider's filled wells in custom-made assay plates, topped with a layer of paraffin oil (Sigma-Aldrich). One MpT was wrapped around an insect pin, while the other MpT remained in the well; fluid droplets accumulating at the ureter (in paraffin oil) were collected at 60 min intervals, from which droplet volume, and thus secretion rate, could be calculated.

### Live imaging of intact renal tubules
For live imaging, animals (of appropriate age/genotype, 7 day old *OregonR* males unless otherwise stated) were rapidly dissected in Schneider's medium (Sigma-Aldrich, S0146) before transferring to glass slides. For staining, dissected tubules were incubated with: JC-1 (5 μg/ml, Abcam, ab113850), BioTracker ATP-Red (10 μM, Millipore, SCT045), MitoTracker Deep Red (100 nM, Invitrogen, M22426), Mito-View 405 (100 nM, BioTium, 70070-T), WGA 633 (1:200, Invitrogen, W21404), DAPI (1:200, Invitrogen, D1306), Phalloidin (1:200, Invitrogen), BODIPY C11 (1:500, Invitrogen, D3861), BODIPY C12 (1:250, Invitrogen, D3822), LipidTox Deep Red (1:200, Invitrogen, H34477), MitoSox (1:250, Invitrogen, M36008), ThiolTracker Violet (1:1000, Invitrogen, T10095), Amplite Probe (4:10, ATT Bioquest, 13806), 2-NBDG (100 μM, Invitrogen, N13195) or LysoTracker Deep Red (1:500, Invitrogen, L12492) for 8–20 min in Schneider's medium (Sigma-Aldrich, S0146) at room temperature in dark conditions, before washing and mounting in Schneider's medium for imaging. Confocal imaging was performed on a Leica TCS SP8 confocal microscope. For antibody stains, tubules were fixed in 4% formaldyhyde (in PBS) for 15 min, washed in PBS-TX-BSA and then incubated overnight with primary antibody (H3K9me3, anti-rabbit primary, Abcam 8898, 1:200) at 4 °C. Tubules were then incubated with secondary antibody (Alexa-488, anti-rabbit, Jackson Immuno Research) for 1 h at room temperature, followed by washes and mounting. For appropriate experiments, tubules were treated with inhibitors: 2-DG (2-deoxy-d-glucose, 50 mM, Sigma, D8375), Sodium Oxamate (Cayman Chemical, 50 mM, 19057) and Rotenone (Sigma, 5 μM, R8875) (in Schneiders medium) for 30 min at room temperature. For imaging of *R2R4 > Laconic*, 7 day male guts were dissected and mounted on a Concanavalin A (Sigma) treated imaging dish in Schneiders medium. Samples were imaged on a Leica SP8 confocal microscope according to published protocols[11] (mTFP 570–522 nm, Venus 532–627 with 405 nm excitation); an average ratio (mTFP/Venus) of 3x ROIs per gut were taken and values normalised to pre-pharmacological treatment levels.

### Senescence associated β-Gal Staining
Tubules were fixed in 2% formaldehyde (in PBS) for 15 min at room temperature, followed by a 1% BSA (in PBS) wash. Tubules were incubated in Cell Event Senescence Green (ThermoFisher, C10850) working solution (1:1000, manufacturers recommended dilution) for 2 h at 37 °C. Samples were then moved to 4 °C overnight, then washed in PBS before mounting and imaging.

### Image processing, analysis and quantification
Briefly, in ImageJ, confocal images were thresholded to segment signal (e.g., peroxisome SKL-GFP), the same was done for the area of the cell.

Thresholded signal area was then measured. Area of signal was divided by total cell/tissue area. For quantification of LD-Peroxisome contact, CapaR>SKL-GFP MpTs were stained with LipidTox and imaged. Number of LDs were counted and noted whether they were in contact with a peroxisome within the z-stack. For 3D Rendering of peroxisome-lipid droplet contact, confocal images were processed in ImageJ. Briefly, z stacks had background subtracted, were resliced then rendered in 3D script (https://doi.org/10.1038/s41592-019-0359-1).

### Analysis of Cell-type and tissue-specific expression from publicly available RNA-Seq data
Cell type-specific expression of target genes in renal PCs and SCs were analysed using published Fly Cell Atlas single nucleus transcriptomic data[19,26] (https://www.flyrnai.org/scRNA/kidney/), generated by dissecting 300 tubules from 5-day-old adult flies, giving expression data for 12,166 total renal tubule nuclei. Seurat-based analysis implements the global-scaling normalisation method "LogNormalize" that normalises the feature expression measurements for each cell by the total expression, then multiplies this by a scale factor (10,000 by default), and log-transforms the result. Post-Seurat expression values for specific genes in 2146 main segment PCs and 1730 main segment SCs were exported and visualised in R. A distinct characteristic of scRNA-seq data is the vast proportion of zeros not seen in bulk RNA-seq data[92]. A 'zero' value reflects a zero expression measurement for that gene in a particular cell, which could either be biological (i.e., lack of gene expression) or non-biological (introduced during sample preparation or due to limited sequencing depths). To compare gene expression between cell types, non-parametric statistical analysis was performed via Wilcoxon rank sum test followed by FDR correction[90]. Tissue specific expression of target genes from the Fly Cell Atlas (https://doi.org/10.1126/science.abk2432) was obtained from flybase.com.

### Biochemical measurements of metabolites
ATP assays: renal tubules dissected from 30 males in Schneiders medium were transferred to 50 μl ice-cold Schneiders. Tissues were ultrasonicated, boiled at 100 °C for 10 min and spun at 5000 g for 5 min. The supernatant was flash frozen in liquid nitrogen and stored at −80 °C until quantification using luciferase-based ATP determination kit (Thermo Scientific A22066) following the manufacturers standard protocol. The ATP extraction protocol was adapted from published protocols[93,94].

TAG assays: renal tubules dissected from 20 male flies in Schneiders medium were transferred to 50 μl ice-cold PBS. Tissues were ultrasonicated, heated at 70 °C for 5 min and then spun at 5000 g for 5 min. The supernatant was flash frozen in liquid nitrogen and stored at −80 °C until quantification using Infinity Triglyceride Reagent (Thermo Scientific TR22421) using a Protocol adapted from the Tartar labs (Brown University).

### Quantitative proteomics of aged tubules
Tubules were dissected from 7 and 28 day adults in Schneider's medium, then transferred via a pulled glass needle into 50 μl ice cold RIPA lysis and extraction buffer (Thermo Scientific 89,900) with Halt Protease and Phosphatase Inhibitor Cocktail (1:100, Thermo Scientific 78,440). Dissected tissues were processed via ultrasonication. Settings were optimised to limit heating of the sample (1 s pulse, 45 s interval, 50% amplitude, 10 s total sonication duration). Lysed samples were kept on ice for 10 min before centrifugation (30 min, 16,430 g, 4 °C) to pellet cellular debris. Supernatant was stored at −80 °C until further use. Protein content of tissue extracts was determined using Micro BCA Protein Assay Kit (Thermo Scientific 23,235) following the manufacturers standard protocol. Each sample of protein extract was collected in triplicate and volumes containing at least 110 μg of protein per replicate were given to the Bristol Proteomics Facility for analysis. The samples were digested with trypsin and the resulting peptides

labelled with Tandem Mass Tag (TMT) eleven plex reagents. The labelled samples were pooled, fractionated by high pH RP chromatography and resulting fractions were analysed by nano-LC MSMS using an SPS-MS3 approach on an Orbitrap Fusion Lumos Mass Spectrometer (Thermo Scientific). The raw data was then analysed using the Proteome Discoverer software *V2.1* (Thermo Scientific) and searched against the UniProt *"Drosophila melanogaster"* databases as well as a common contaminants database.

## Data analysis and visualisation
All statistical analysis and data visualisation were performed in an R computing environment, with Tidyverse[95] packages, using rStudio. The statistical tests used for each experiment have been specified in the figure legends. Figures were compiled using Adobe Illustrator (Adobe, California).

## Reporting summary
Further information on research design is available in the Nature Portfolio Reporting Summary linked to this article.

## Data availability
The minimum dataset necessary to interpret, verify and extend the research in this article is accessible within the manuscript and its Supplementary Information. For graphical display items throughout the study, raw data points have been plotted overlaid on each box-and-whisker plot to maximise the reproducibility of the research data. Fly Cell Atlas single nuclear transcriptomic data is publicly available at https://www.flyrnai.org/scRNA/kidney/ and https://flycellatlas.org/scope. Fly Cell Atlas raw snRNA-seq reads are available in the Gene Expression Omnibus (GEO) database, https://www.ncbi.nlm.nih.gov/geo (accession no. GSE202575). UniProt *"Drosophila melanogaster"* databases are publicly available at https://www.uniprot.org/proteomes/UP000000803. The mass spectrometry data have been deposited to the ProteomeXchange Consortium via the PRIDE partner repository with the dataset identifier PXD047350. Raw source data are also provided with this paper. Source data are provided with this paper.

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

## Acknowledgements

We would like to thank members of the Weavers, Martin and Richardson labs for helpful discussion. We also thank the Wolfson Bioimaging Facility (University of Bristol), Bristol Proteomics Facility (University of Bristol), Bloomington Stock Centre (University of Indiana), Vienna *Drosophila* Resource Centre, Barry Denholm (Edinburgh), Julian Dow (Glasgow), Hyung Don Ryoo (NYU), Ioannis Trougakos (Athens) and Irene Miguel-Aliaga (Imperial College London) for *Drosophila*. This research was funded in part by the Wellcome Trust a Wellcome Trust and Royal Society Sir Henry Dale Fellowship to H.W. [208762/Z/17/Z] and a Sir Jules Thorn PhD Scholarship (in partnership with the MRC GW4 DTP) to J.H. For the purpose of Open Access, the author has applied a CC BY public copyright license to any Author Accepted Manuscript arising from this submission.

## Author contributions

J.H. performed the experiments and, together with H.W., conceived the study and wrote the manuscript.

## Competing interests

The authors declare no competing interests.
