## [Peer Review File · Nature Communications]

Functional-metabolic coupling in distinct renal cell types
coordinates organ-wide physiology and delays premature
ageingREVIEWER COMMENTS

Reviewer #1 (Remarks to the Author):

This is an interesting and multidimensional paper, which greatly enhances our understanding of metabolic homeostasis in a vital tissue.

Major comments:

To my mind, the biggest weakness of this MS is that relies heavily on microscopy, and attempts to quantify reporter genes and fluorescent probes, rather than direct biochemical measurements of the inferred metabolites which would have given direct, quantitative data on a linear scale. The story is, however, plausible; and I am inclined to let it go.

Minor comments:

when making the case for the tubule as an essential tissue, it may be worth mentioning that it's the only tissue which does not remodel during pupation.

Results: figure 1. the mitochondrial density in tubules is (perhaps more) nicely shown in DOI: 10.1074/jbc.M603002200, fig. 3A-E, with mitochondria all through the cells, especially the principal cells - and every microvillus containing a mitochondrion (as shown by em in *Calliphora* by Bradley). Also critical to include here, this paper directly assesses mitochondrial potential with the voltage-sensitive dye JC-1, and show that blockade of V-ATPase with bafilomycin nearly doubles cellular [ATP], confirming the V-ATPase as the largest single energy sink in the tubule (Fig. 5). On that note, it would have been preferable to have used a direct determination of [ATP] in this paper, rather than the less precise ATP-red fluorescence.

Fig. 1. What is the basis for asserting that MalA1-GFP, PGI-GFP, and LDH-GFP accurately and fully represent the expression of the named genes? Given that such reporter constructs only rarely do so, it would be nice to add a caveat to the text along the lines "While it cannot be asserted that any individual reporter construct will completely and faithfully map the expression of its cognate gene, the impression given by a related panel of such reporters is that..."

Methods: Ramsay assays. Maddrell always asserted that droplets should be sampled as frequently as possible (he worked to 10 min) to allow dynamic aspects of secretion to be studied. This is less important here; but given that several of the neuropeptide modulators

of tubule are also thought to impact metabolism, it would have been interesting to see hormonal impact on these metabolic pathways.

Results 185: It's worth going back at least as far as Wessing to acknowledge that stellate (type II) cells are thought to be much less metabolically active than principal.

Results line 196: it is welcome to see a detailed analysis of peroxisomal function in this tissue. The authors are presumably focussing on the main segment of the tubule for this analysis (this should be made clear); it may be worth mentioning that the peroxisomes of the initial segment appear to have been co-opted to a different role, in calcium storage excretion (10.1152/physiolgenomics.00038.2006).

Results, line 326: ERR. This section is somewhat speculative (and the results somewhat confusing), but I guess it is marked as such.

Reviewer #2 (Remarks to the Author):

Dear Editor / Authors.

The paper addresses a long-standing conundrum in the field of biochemistry and metabolism: how do different cell types meet their differing metabolic needs? Additionally, how does a cell adapt its metabolic 'flux' in response to different stressor and at differing life history stages? These are important questions that affects our basic understanding of tissue and organ biology. It is thought that the mechanisms underpinning the heterogeneity of metabolic needs, might be therapeutic targets for the promotion of healthy ageing.

The team nicely addresses the question using the *Drosophila* renal organ known as the Malpighian tubule. The tubules regulate ion and water balance in insects and comprise a small number of different and well-characterised cell types, most notably the Principal and Stellate Cells. The model is supported by established cell-specific genetic tools, genetic reporters and it is highly amenable to pharmacological intervention. *Drosophila* is also well known as a useful model with which to study ageing in complex organisms and the renal tubules reflect mechanisms that are conserved between insects and mammals. The team also use publicly available single cell gene expression data to support their reasoning and hypotheses.

The team find that Principal and Stellate cells within a tissue differentially couple their energy needs with functional outputs. In short, the tissue consists of different metabolic compartments. The team suggest this might be to avoid competition between differing cells for 'precious' substrates. They show that glucose is either transported into or generated within principal cells and then directed to the pentose phosphate pathway, rather than through latter stages of glycolysis but that the majority of ATP is generated by fatty acid oxidation. They also show that tubule function declines as a function of ageing and that concomitant changes to a transcription factor (ERR) may help explain this ageing phenotype.

Overall the data are compelling especially in relation to fatty acid oxidation by PCs. To find that two different cell types within a tissue have differing metabolic pathways is perhaps not surprising but the team's overarching hypothesis is interesting – i.e. this compartmentalisation of metabolism has evolved as a necessity, it avoid adjacent yet different cell types competing with each other. Whether this is a general phenomenon is not clear. The team have done a great job addressing what at first appears to be a highly intractable problem and I'd recommend publishing this article after revision.

I have made a lot of suggestions that I think cause a lot of editorial work (mainly adjusting text and figures). There may be some issues with the stats but I doubt these will alter the hypotheses if indeed they need to be corrected. Some additional, yet limited, microscopy work and controls to clarify a few points would be helpful too.

Dear Authors:

Minor:

The abstract and introduction start with comment about tissue health, whereas the discussion ends with a comment about tumour microenvironments. It might be best to focus on healthy ageing and remove reference to tumour biology?

Abstract: Define the abbreviations, PPP, FA, ERR.

Stylistically, there is a reliance on hyperbole that could probably be toned-down, for example p3, 1st sentence of the introduction uses the terms, 'extraordinarily' and 'astoundingly'. Personally, these feel out of place in a technical document and arguably they are somewhat redundant because it is widely accepted that biology is complicated.

Intro general: what is missing here is a commentary on how any single cell type adjusts its metabolism or substrate usage according to different conditions. Yeast would be an ideal focal point, for example, moving it to and from nitrogenous compounds led to autophagy genes being identified. High and low oxygen and HIF1a (I note this comes up later – but it could feature in the intro?). Whilst there are an abundance of mechanisms that cells use to adapt their metabolism, your point seems to be more about how within a tissue these mechanisms are diverse as a necessity – i.e. that there is an evolutionary pressure for differing cell types within a tissue to have dissimilar metabolic mechanisms – it reduces competition for precious substrates. I might be wrong – but this hypothesis perhaps needs a more explicit presentation.

Methods general: The team are not really conducting experiments "in vivo", these are ex vivo methods that rely on dissection and preparation in culture media.

scRNASeq Data: A lot of the graphs display scRNASeq data that is not new to the study, it is from a publicly available source. Firstly, this source needs to be cited (presumably it is from the FlyCell Atlas?). Secondly, I'd argue that these graphs should be removed from the primary figures and presented as a supplemental table. Sample size needs stated (is it 10,000 cells from a single tissue sample or is it the mean of 3-4 sets of 10,000 cells etc..?). The stats needs to be considered too – these data do not look normally distributed and in some cases the data is clearly bimodal, yet parametric tests have been used to compare means. A lot of conclusions rest on the two cell types having identical expression levels for internal control genes and it is not clear which genes were used as controls – for example was it GAPDH? Can it be safely assumed the levels of control gene expression are identical between the two cell types? This is important because many of the differences are relatively small. Also, a lot of hypotheses are being tested and there should be some sort of correction

(e.g. perhaps a Bonferroni correction?) applied to the testing strategy.

Minor specifics:

L30. There is the statement, “This vital life-sustaining organ....”, the authors will know that a human can live with a single kidney so it might be better to say ‘the kidneys provide vital..’ or ‘kidney function is vital...’.

L33. Remove the word ‘profoundly’, it seems unnecessary in this context. ATP-dependent process are vital.

L33. “Kidneys ... extraordinarily high bioenergetic ... rate of 400kcal per kg per day”. This statement would benefit from a little more context – the authors could provide comparative data from that paper for the brain, liver and heart etc..., in order to emphasise their claim.

L80. “In this study, we demonstrate strategic metabolic partitioning between distinct renal cell types not only supports their unique functional roles but sustains tissue-wide excretory activity and delays 81 premature ageing”. Insert ‘that’ between ‘demonstrate / strategic’? And inset ‘also’ between ‘but / sustains’?

L86. Define ETC as Electron Transport Chain (it does not seem to be defined in the paper).

L93. OXPHOS is used for the first time but it is defined on L108 – can you go through the document to ensure abbreviations are defined at first use please?

L98. “their dramatic disruption’, dramatic seems unnecessary – perhaps too dramatic.

L100. Are you suggesting a general hypothesis regarding tissue complexity where different cell types may need to employ different substrate-usage mechanisms because this is less competitive and toxic compared to the use of a single common substrate / metabolite production pathway? That’s a really interesting idea – however it might need articulated

more clearly (and are there any other examples?)?

Figure 1H. State the values in the results section because it is not clear what the relative difference is for Glut1 between PCs and SCs.

L138. “revealed significantly increased uptake...” should read ‘revealed significantly greater uptake relative to’.

L144. “In contrast, inhibition of glucose import by SCs had little effect on tubule function under basal conditions (Figure 1O)” This refers to Figure 1N and 1O. Why do you assume that knock down of the same target gene (Glut1) with two different drivers has been equivalent? Do you know that knock down of Glut1 in SCs was successful and if so to what level?

L150 – I’m not sure you should say this is ‘surprising’, it doesn’t seem to be. The levels of ATP do not decrease because ATP is not being used at the same rate – largely because secretion is reduced. You have reduced glucose transport, which (conceivably) might lead to reduced ATP only if the secretion rate was maintained at control levels.

Figure Q and Q’. Are these figures adjusted for total cell volume? The cells are very different total volumes and therefore the total amount of enzyme activity is relevant here – not the amount quantified in a confocal slice. The cells are likely to have very different total enzyme activity.

I know it seems unlikely to change any interpretation of the data but in Fig 1 the authors are using a lack of an effect as evidence for an alternative hypothesis, so it is important to add positive controls as follows (they can argue otherwise):

Figure 1S – you need to show a positive control for PGI RNAi (a cell type where you knock it down to good effect and see an effect on readouts of glycolysis).

Figure 1T – you need to show a positive control for E4P.

Figure 1V – you need to show a positive control for Oxamate.

L162 “If PCs do not rely on glucose metabolism to meet their energetic demands...”, this seems like an oversimplification of the data – the cells do rely on glucose. They have glucose transporters, enzymatic mechanisms to make glucose and their function is sensitive to inhibition by 2DG. The statement probably needs revised a little.

L168. Re-phrase ‘vast quantities’ and just state the fold difference between the metabolic pathways that generate ATP.

L306-319: this is for the discussion.

L352-360: seems like discussion / speculation.

Figure 7A. Is this based on immunofluorescence of nuclear ERR? This is complicated because the nucleus of older PCs is larger than in younger cells, is it not? This might ‘dilute’ an ERR nuclear signal. Is there a way to quantify the entire ERR fluorescence in a nucleus of young and older PCs? This seems important given the conclusion is that the ERR protein levels are declining as a consequence of ageing.

L432, “we reveal strict functional-metabolic”, insert ‘that’ between reveal / strict?

Methods:

L19. It is conspicuous that there is no mention of bicarbonate being added to the Schneider’s media. Can the team confirm whether bicarbonate was added to Schneider’s Medium?

JC-1 is used. Monochrome images are shown in Fig.S1 and these are red. It is typical for JC-1 data to show the dual orange-green fluorescence of the probe and present a ratio of orange-green (aggregates-monomers). The team need to explain how JC-1 was imaged and interpreted. It would be important to show the dual fluorescence, if possible.

Could the authors state when images are single confocal sections or z-stacks please?

Statistics.

The use of the snRNAseq data: a lot of graphs show bi-modal distribution of data – with many points close of zero and others at higher values with no continuum between – for example in Fig 4 all the data from SCs looks like this. The reason for all the ‘zero’ data needs to be addressed. It also looks like the data are non-normally distributed and from the methods it looks like parametric tests are used throughout.

For example, in Supplemental Figure 1 the expression data need n numbers stated and there needs to be justification of using parametric tests because all the data look skewed and non-normally distributed. Also, there are some means that are almost identical, yet the stats results show a P value $\ll 0.01$ (e.g. Complex II SdhB, 0.18 vs 0.17). This starts to look like an over-sampling error / false positive.

You also need at some point to define the source of the scRNASeq data, there doesn't seem to be a reference cited.

Also, what internal control gene is used and is this equivalent between cell types – for example, if it is actin or b-actin then SCs would be expected to have a greater number of transcripts per cell than PCs. Also, PCs have a huge nucleus compared to SCs – how does this affect the interpretation of the data? A lot of transcripts will be in the rough ER – are these taken into account.

Which data in Figure 4 were tested with an ANOVA?

RE: Manuscript NCOMMS-23-22648

In summary, we have strengthened our description of renal metabolism by complementing our microscopy-based analyses with quantitative biochemical measurements of specific metabolites (including ATP and lipids). We have validated that our genetic and pharmacological methods to inhibit late steps of glycolysis are effective by performing additional analyses using the glycolytic biosensor Laconic. We have provided additional genetic evidence to support our suggestion that Glut1 is primarily required in PCs, but not SCs, for robust tubule secretion. We have now included additional microscopy data to demonstrate how mitochondrial density and mitochondrial activity are enriched in PCs relative to SCs. We have also carried out improved, non-parametric analyses of the published Fly Cell Atlas snRNA-seq data and included additional analyses of 'housekeeping' or 'control' genes within the different renal cell types. Finally, as suggested by both Reviewers, we have made extensive improvements to the manuscript text to improve clarity, remove unnecessary hyperbole and included additional important references. We are grateful for the reviewers' suggestions that led us to perform these additional experiments and quantification, which we feel have greatly improved the paper.

The text below provides a detailed response to individual reviewer comments with our response to each suggestion/comment highlighted in bold following the text of their reviews. We have also highlighted each individual revision within the updated manuscript in yellow.

Reviewer #1:

This is an interesting and multidimensional paper, which greatly enhances our understanding of metabolic homeostasis in a vital tissue.

Major comments:

To my mind, the biggest weakness of this MS is that relies heavily on microscopy, and attempts to quantify reporter genes and fluorescent probes, rather than direct biochemical measurements of the inferred metabolites which would have given direct, quantitative data on a linear scale. The story is, however, plausible; and I am inclined to let it go.

We thank the Reviewer for this suggestion. We have now performed direct biochemical measurement of key metabolites (ATP and lipids) to provide additional quantitative data to complement our original microscopy-based analysis. For this, we adapted published assays (e.g. as in Terhzaz et al. 2010 *Physiol Genomics*). These new data are shown in Figure 1 ([ATP] following PC>*glut1-RNAi*), Figure S1 ([ATP] following rotenone treatment) and Figure S2 (both [ATP] and [lipid] following PC>*dCPT1-RNAi*). Our reason for primarily using microscopy-based measurement of fluorescent probes and reporters was to enable cell-type specific analysis of metabolite levels. Given the two main constituent cell types (PCs and SCs) are intermingled along the length of the tubule, it is challenging to separate these cell types for biochemical assays whilst faithfully retaining their native metabolic state. For a comprehensive quantification of cell type-specific metabolites, we would ideally perform single cell metabolomics, but feel this ambitious experiment is beyond the scope of the current manuscript and should form the focus of a future study.

Minor comments:

when making the case for the tubule as an essential tissue, it may be worth mentioning that it's the only tissue which does not remodel during pupation.

We agree this is an interesting point and have changed the text accordingly (page 4).

Results: figure 1. the mitochondrial density in tubules is (perhaps more) nicely shown in DOI: 10.1074/jbc.M603002200, fig. 3A-E, with mitochondria all through the cells, sepecially the principal cells - and every microvillus containing a mitochondrion (as shown by em in Calliphora by Bradley). Also critical to include here, this paper directly assesses mitochondrial potential with the voltage-sensitive dye JC-1, and show that blockade of V-ATPase with bafilomycin nearly doubles cellular [ATP], confirming the V-ATPase as the largest single energy sink in the tubule (Fig. 5).

We apologise for not originally including this insightful reference (Terhzaz et al 2006), which provides important insight into mitochondrial distribution, activity and V-ATPase activity within tubule cells. We have now referred to this study in more detail in the text (page 6) and also included improved images of tubule cell mitochondria (Figure S1B-C).

On that note, it would have been preferable to have used a direct determination of [ATP] in this paper, rather than the less precise ATP-red fluorescence.

As described above in our response to Major Comment #1, we have now assessed ATP levels more directly using biochemical assays (see Fig. 1, Fig. S1 and Fig. S2).

Fig. 1. What is the basis for asserting that MalA1-GFP, PGI-GFP, and LDH-GFP accurately and fully represent the expression of the named genes? Given that such reporter constructs only rarely do so, it would be nice to add a caveat to the text along the lines "While it cannot be asserted that any individual reporter construct will completely and faithfully map the expression of its cognate gene, the impression given by a related panel of such reporters is that..."

We are grateful for this suggestion and have now included this caveat in the text on page 7.

Methods: Ramsay assays. Maddrell always asserted that droplets should be sampled as frequently as possible (he worked to 10 min) to allow dynamic aspects of secretion to be studied. This is less important here; but given that several of the neuropeptide modulators of tubule are also thought to impact metabolism, it would have been interesting to see hormonal impact on these metabolic pathways.

We agree with the Reviewer that exploring whether challenging the tubules (away from homeostasis) would impact PC or SC metabolism, however we envision these analyses are beyond the scope of the current study and should form the basis of a future comprehensive study.

Results 185: It's worth going back at least as far as Wessing to acknowledge that stellate (type II) cells are thought to be much less metabolically active than principal.

We thank the reviewer for this suggestion and have now included additional references from Wessing on page 6.

Results line 196: it is welcome to see a detailed analysis of peroxisomal function in this tissue. The authors are presumably focussing on the main segment of the tubule for this analysis (this should be made clear); it may be worth mentioning that the peroxisomes of the initial segment appear to have been co-opted to a different role, in calcium storage excretion (10.1152/physiolgenomics.00038.2006).

We apologise for the confusion and have now made it clear that we are focusing on the tubule's main segment for our analyses (text page 9). We have also included this interesting reference on peroxisome function in the initial segment (page 10).

Results, line 326: ERR. This section is somewhat speculative (and the results somewhat confusing), but I guess it is marked as such.

As the reviewer suggests, this section of the manuscript exploring the role of ERR on programming tubule metabolism (Figure 6) is more speculative than other sections of the manuscript. Nevertheless, we feel that the presented ERR data brings us an important step closer to understanding how distinct metabolic profiles are established and/or maintained in different cell types within a single (renal) tissue. We have now tried to bring more clarity to the text accompanying the ERR results in Figure 6.

Reviewer #2

Overall the data are compelling especially in relation to fatty acid oxidation by PCs. To find that two different cell types within a tissue have differing metabolic pathways is perhaps not surprising but the team's overarching hypothesis is interesting – i.e. this compartmentalisation of metabolism has evolved as a necessity, it avoid adjacent yet different cell types competing with each other. Whether this is a general phenomenon is not clear. The team have done a great job addressing what at first appears to be a highly intractable problem and I'd recommend publishing this article after revision. I have made a lot of suggestions that I think cause a lot of editorial work (mainly adjusting text and figures). There may be some issues with the stats but I doubt these will alter the hypotheses if indeed they need to be corrected. Some additional, yet limited, microscopy work and controls to clarify a few points would be helpful too.

We thank the Reviewer for their overall positivity; we have now addressed each of your specific concerns as outlined below.

Minor:

The abstract and introduction start with comment about tissue health, whereas the discussion ends with a comment about tumour microenvironments. It might be best to focus on healthy ageing and remove reference to tumour biology?

We apologise for the confusion here; we agree with the Reviewer's suggestion and have now moved the text referring to tumour biology.

Abstract: Define the abbreviations, PPP, FA, ERR.

We have now defined these abbreviations in the Introduction on page 5.

Stylistically, there is a reliance on hyperbole that could probably be toned-down, for example p3, 1st sentence of the introduction uses the terms, 'extraordinarily' and 'astoundingly'. Personally, these feel out of place in a technical document and arguably they are somewhat redundant because it is widely accepted that biology is complicated.

We apologise for our over-use of hyperbole and have toned-down the language accordingly; for example as suggested, we have removed 'astoundingly' and 'extraordinarily' from the first paragraph of the Introduction.

Intro general: what is missing here is a commentary on how any single cell type adjusts its metabolism or substrate usage according to different conditions. Yeast would be an ideal focal point, for example, moving it to and from nitrogenous compounds led to autophagy genes being identified. High and low oxygen and HIF1a (I note this comes up later – but it could feature in the intro?). Whilst there are an abundance of mechanisms that cells use to adapt their metabolism, your point seems to be more about how within a tissue these mechanisms are diverse as a necessity – i.e. that there is an evolutionary pressure for differing cell types within a tissue to have dissimilar metabolic mechanisms – it reduces competition for precious substrates. I might be wrong – but this hypothesis perhaps needs a more explicit presentation.

We thank the Reviewer for this suggestion and have now expanded our discussion of these points in the first paragraph of the Introduction.

Methods general: The team are not really conducting experiments "in vivo", these are ex vivo methods that rely on dissection and preparation in culture media.

We agree with the Reviewer and have modified the text accordingly.

scRNASeq Data: A lot of the graphs display scRNASeq data that is not new to the study, it is from a publicly available source. Firstly, this source needs to be cited (presumably it is from the FlyCell Atlas?).

We apologise for this omission; we have now cited the Fly Cell Atlas each time it is used to compare PC vs SC gene expression.

Secondly, I'd argue that these graphs should be removed from the primary figures and presented as a supplemental table. Sample size needs stated (is it 10,000 cells from a single tissue sample or is it the mean of 3-4 sets of 10,000 cells etc..?). The stats needs to be considered too – these data do not look normally distributed and in some cases the data is clearly bimodal, yet parametric tests have been used to compare means. A lot of conclusions rest on the two cell types having identical expression levels for internal control genes and it is not clear which genes were used as controls – for example was it GAPDH? Can it be safely assumed the levels of control gene expression are identical between the two cell types? This is important because many of the differences are relatively small. Also, a lot of hypotheses are being tested and there should be some sort of correction (e.g. perhaps a Bonferroni correction?) applied to the testing strategy.

We apologise for the confusion here. We have now included more details of the sample sizes used to generate the Fly Cell Atlas data in the relevant Methods section. As the Reviewer suggests, the snRNAseq data are not normally distributed (they are rather bimodal). We have sought advice from groups that routinely perform scRNAseq, and following their advice, we have now performed non-parametric statistical analyses (Wilcoxon rank sum) with a multiple testing (FDR) correction. Regarding the use of control genes, we have now included additional analysis (Figure S1A) to show that i) traditional PC (e.g. *CapaR*, *Uro*, *Cut*, *DH31-R*) and SC (*Tsh*, *SecCI*, *Drip* and *Prip*) cell type-specific markers are enriched in PCs and SCs, respectively within the snRNAseq data and ii) control genes (e.g. *Gapdh1* and *RpS20*) are expressed at similar levels within PCs and SCs.

Minor specifics:

L30. There is the statement, “This vital life-sustaining organ...”, the authors will know that a human can live with a single kidney so it might be better to say ‘the kidneys provide vital..’ or ‘kidney function is vital...’.

We apologise for the confusion here and have now modified the text accordingly.

L33. Remove the word ‘profoundly’, it seems unnecessary in this context. ATP-dependent process are vital.

We have now modified the text accordingly.

L33. “Kidneys ... extraordinarily high bioenergetic ... rate of 400kcal per kg per day”. This statement would benefit from a little more context – the authors could provide comparative data from that paper for the brain, liver and heart etc..., in order to emphasise their claim.

We thank the reviewer for this suggestion and have now included these comparative data on page 3.

L80. “In this study, we demonstrate strategic metabolic partitioning between distinct renal cell types not only supports their unique functional roles but sustains tissue-wide excretory activity and delays 81 premature ageing”. Insert ‘that’ between ‘demonstrate / strategic’? And inset ‘also’ between ‘but / sustains’?

We have now modified the text accordingly.

L86. Define ETC as Electron Transport Chain (it does not seem to be defined in the paper).

We have now modified the text accordingly.

L93. OXPHOS is used for the first time but it is defined on L108 – can you go through the document to ensure abbreviations are defined at first use please?

We apologise for these omissions, we have now modified the text accordingly.

L98. “their dramatic disruption’, dramatic seems unnecessary – perhaps too dramatic.

We have now modified the text accordingly.

L100. Are you suggesting a general hypothesis regarding tissue complexity where different cell types may need to employ different substrate-usage mechanisms because this is less competitive and toxic compared to the use of a single common substrate / metabolite production pathway? That’s a really interesting idea – however it might need articulated more clearly (and are there any other examples?)?

We agree with the Reviewer, this is the hypothesis we were suggesting. We have now modified the text (page 5) to articulate this idea more clearly.

Figure 1H. State the values in the results section because it is not clear what the relative difference is for Glut1 between PCs and SCs.

We apologise for the lack of clarity here; we have now explicitly stated the mean expression values for each of the genes on the graph.

L138. “revealed significantly increased uptake...” should read ‘revealed significantly greater uptake relative to’.

We have now modified the text accordingly.

L144. “In contrast, inhibition of glucose import by SCs had little effect on tubule function under basal conditions (Figure 1O)” This refers to Figure 1N and 1O. Why do you assume that knock down of the same target gene (Glut1) with two different drivers has been equivalent? Do you know that knock down of Glut1 in SCs was successful and if so to what level?

To complement our original data, we have now demonstrated that expression of *glut1-RNAi* throughout the tubule in both PCs and SCs (using the ubiquitous *Act5c-gal4* driver) leads to a similar reduction in tubule secretion to that of PC-driven *glut1-RNAi* alone (Figure S1I). We have also demonstrated that PC and SC Gal4 drivers give rise to similar expression levels of a UAS-driven GFP construct (Figure S1H); in fact the SC Gal4 is more efficient at driving GFP expression than the PC Gal4. Nevertheless, we agree with the Reviewer that without performing RT-qPCR on isolated PCs and SCs (where PC and SC separation would be technically challenging), we can not be sure that we are achieving exactly the same level of knockdown of *glut1* in PCs and SCs; we have therefore modified the text to be more cautious in our interpretation of the data.

L150 – I’m not sure you should say this is ‘surprising’, it doesn’t seem to be. The levels of ATP do not decrease because ATP is not being used at the same rate – largely because secretion is reduced. You have reduced glucose transport, which (conceivably) might lead to reduced ATP only if the secretion rate was maintained at control levels.

We have now modified the text accordingly.

Figure Q and Q’. Are these figures adjusted for total cell volume? The cells are very different total volumes and therefore the total amount of enzyme activity is relevant here – not the amount quantified in a confocal slice. The cells are likely to have very different total enzyme activity.

The Reviewer has made a very valid point; the original data measured PGI-GFP and LDH-GFP intensity per unit area of the cell. We have now measured total intensity across the cell, to indicate how total enzyme levels might vary between PCs and SCs (Figure S1J-K); as the Reviewer suggests, given the considerably larger size of PCs relative to SCs our new analyses indicate that PCs possess higher total PGI levels than SCs, although the total LDH (based on the GFP reporter) is not different between the two cell types.

I know it seems unlikely to change any interpretation of the data but in Fig 1 the authors are using a lack of an effect as evidence for an alternative hypothesis, so it is important to add positive controls as follows (they can argue otherwise):

- Figure 1S – you need to show a positive control for PGI RNAi (a cell type where you knock it down to good effect and see an effect on readouts of glycolysis).

We have now demonstrated that PGI-RNAi expression within enterocytes within the fly gut (using the *R2R4-gal4* driver) leads to significant drop in lactate (Figure S1L), as inferred using the genetically-encoded FRET lactate biosensor Laconic (using established assays as in Hudry et al., 2019 and Gonzalez-Gutierrez et al., 2019).

- Figure 1T – you need to show a positive control for E4P.

We have now assessed whether E4P treatment of enterocytes leads to a significant drop in lactate, as inferred using the genetically-encoded biosensor Laconic. However, our data suggest that E4P treatment does not lead to a strong reduction in lactate production (see graph below) compared to that we observed using PGI-RNAi and Oxamate treatment; for this reason, we envision it may be appropriate to remove the original E4P data from Figure 1. Nevertheless, given that we show PGI-RNAi and oxamate treatment are successful at markedly reducing lactate levels in a tissue in which they are required (enterocytes, in a manner comparable to published data; Hudry et al., 2019 and Gonzalez-Gutierrez et al., 2019), we envision our suggestion that late stage glycolytic enzymes are not essential in PCs for robust tubule secretion remains valid.

- Figure 1V – you need to show a positive control for Oxamate.

We have now demonstrated that Oxamate treatment of fly enterocytes (under equivalent conditions to our renal tubule experiment) also leads to a significant drop in lactate (Figure S1M), as inferred using the glycolytic biosensor Laconic (using established assays as in Hudry et al., 2019 and Gonzalez-Gutierrez et al., 2019).

L162 “If PCs do not rely on glucose metabolism to meet their energetic demands...”, this seems like an oversimplification of the data – the cells do rely on glucose. They have glucose transporters, enzymatic mechanisms to make glucose and their function is sensitive to inhibition by 2DG. The statement probably needs revised a little.

We have now modified the text accordingly.

L168. Re-phrase ‘vast quantities’ and just the state fold difference between the metabolic pathways that generate ATP.

We have now modified the text accordingly.

L306-319: this is for the discussion.

We have now modified the text accordingly.

L352-360: seems like discussion / speculation.

We have now modified this section of the text.

Figure 7A. Is this based on immunofluorescence of nuclear ERR? This is complicated because the nucleus of older PCs is larger than in younger cells, is it not? This might ‘dilute’ an ERR nuclear signal.

Is there a way to quantify the entire ERR fluorescence in a nucleus of young and older PCs? This seems important given the conclusion is that the ERR protein levels are declining as a consequence of ageing. **The data on ERR levels in Figure 7A was generated using mass spectrometry on protein extracted from intact, whole renal tubules; we have included detailed protocols on the generation of this data in the Methods and made the raw data available online. In the text accompanying these data in Figure 7A we have made it clear that ERR levels are obtained from young and old whole tubules, rather than PCs.**

For completeness, we have also quantified nuclear size in young (7-day) and aged (35-day) PCs and we find no detectable difference (data shown below).

L432, “we reveal strict functional-metabolic”, insert ‘that’ between reveal / strict?
We have now modified the text accordingly.

Methods:

L19. It is conspicuous that there is no mention of bicarbonate being added to the Schneider’s media. Can the team confirm whether bicarbonate was added to Schneider’s Medium?

We use Schneider’s Medium S0146 from Sigma-Aldrich, which includes NaHCO₃ (0.4 g/L), glucose: 2 g/L (Dextro) and L-glutamine (1.8 g/L). We have now made reference to this specific Schneider’s Medium in the Methods.

JC-1 is used. Monochrome images are shown in Fig.S1 and these are red. It is typical for JC-1 data to show the dual orange-green fluorescence of the probe and present a ratio of orange-green (aggregates-monomers). The team need to explain how JC-1 was imaged and interpreted. It would be important to show the dual fluoresce, if possible.

As the Reviewer suggests, JC-1 is traditionally imaged in both red/orange and green channels, which indicate relative levels of JC-1 aggregates and monomers, respectively. JC-1 aggregates (red channel) are typically indicative of mitochondrial polarisation and hence mitochondrial activity; in our original manuscript, we thus quantified only the red fluorescence output of JC-1. We have now included dual colour images for JC-1 in wild-type tubules (Figure S1E-E’’) and quantified the ratio of JC-1 aggregates (red) – monomers (green) in PCs and SCs (Figure S1E’’’).

Could the authors state when images are single confocal sections or z-stacks please?

We have now modified the text accordingly.

Statistics:

The use of the snRNAseq data: a lot of graphs show bi-modal distribution of data – with many points close of zero and others at higher values with no continuum between – for example in Fig 4 all the data from SCs looks like this. The reason for all the ‘zero’ data needs to be addressed. It also looks like the data are non-normally distributed and from the methods it looks like parametric tests are used throughout.

We apologise for the confusion here and lack of clarity. As explained in our response above regarding analysis of the Fly Cell Atlas snRNA-seq data, we have now included additional details in the Methods regarding data generation and analysis. The graphs for the scRNA-seq data are plots of 'expression values' (following Seurat-based scaling and normalisation) for that gene in each individual cell (with a unique barcode).

Regarding the 'zero' values: a distinct characteristic of scRNA-seq data is the vast proportion of zeros unseen in bulk RNA-seq data (see Jiang et al 2022 *Genome Biology* for a detailed explanation). Each plotted value represents the expression of that gene from a single cell. Thus a 'zero' value reflects a zero expression measurement for that gene in a particular cell. These zeros are likely to be a mixture of biological zeros and non-biological zeros; while biological zeros likely indicate a lack of gene expression, non-biological zeros represent missing values artificially introduced during the generation of scRNA-seq data (e.g. during the preparation of biological samples for sequencing or due to limited sequencing depths). As the Reviewer suggests, given the vast numbers of zeros, the data is not normally distributed and we have therefore re-analysed the data with more appropriate non-parametric tests (Wilcoxon rank sum with FDR correction, see our response on page 4 above).

For example, in Supplemental Figure 1 the expression data need n numbers stated and there needs to be justification of using parametric tests because all the data look skewed and non-normally distributed. Also, there are some means that are almost identical, yet the stats results show a P value $\ll 0.01$ (e.g. Complex II SdhB, 0.18 vs 0.17). This starts to look like an over-sampling error / false positive.

For all analyses of the Fly Cell Atlas scRNAseq data, we have used expression data from 2146 main segment PCs and 1730 main segment SCs; we now make this point clear in the relevant Methods section. As described above, we have now repeated our statistical analyses using non-parametric tests (Wilcoxon rank sum test) with FDR corrections for the large sample size.

You also need at some point to define the source of the scRNASeq data, there doesn't seem to be a reference cited.

We have now modified the text accordingly.

Also, what internal control gene is used and is this equivalent between cell types – for example, if it is actin or b-actin then SCs would be expected to have a greater number of transcripts per cell than PCs. Also, PCs have a huge nucleus compared to SCs – how does this affect the interpretation of the data? A lot of transcripts will be in the rough ER – are these taken into account.

The Fly Cell Atlas transcriptomic data was generated on isolated nuclei, so will only measure nuclear transcripts (not transcripts in the ER). The current standard method for plotting scRNA-seq expression data does not involve normalisation to a control gene (as is traditional in techniques such as RT-qPCR), rather it is routine to plot the number of detected transcripts per gene per cell. Nevertheless, as described above, we have now included additional expression analyses showing enrichment of 'positive control' genes for PCs and SCs (Figure S1A) as well as uniform expression of internal control genes (e.g. *Gapdh1* and *RpS20*; Figure S1A).

Which data in Figure 4 were tested with an ANOVA?

We apologise for the confusion here, ANOVA was not used to analyse data in Figure 4 and the text has been updated accordingly.

REVIEWERS' COMMENTS

Reviewer #1 (Remarks to the Author):

I thank the authors for their thorough response to my queries/suggestions. I think this is a very interesting and valuable MS in the field, and I have no further issues to raise.

Reviewer #2 (Remarks to the Author):

The authors have thoroughly addressed my comments. This is a well written paper with interesting findings. I think the team explain a very complex aspect of tissue level metabolic partitioning with clarity. The supporting evidence is robust and presented logically.